# Noncanonical roles of ATG5 and membrane atg8ylation in retromer assembly and function

Masroor Ahmad Paddar[1,2], Fulong Wang[1,2], Einar S Trosdal[1,2], Emily Hendrix[3], Yi He[3], Michelle R Salemi[4], Michal Mudd[1,2], Jingyue Jia[1,2], Thabata Duque[1,2], Ruheena Javed[1,2], Brett S Phinney[4], Vojo Deretic[1,2]*

[1]Autophagy, Inflammation and Metabolism Center of Biochemical Research Excellence, University of New Mexico School of Medicine, Albuquerque, United States; [2]Department of Molecular Genetics and Microbiology, University of New Mexico School of Medicine, Albuquerque, United States; [3]Department of Chemistry & Chemical Biology, The University of New Mexico, Albuquerque, United States; [4]Proteomics Core Facility, University of California, Davis, Davis, United States

## eLife Assessment

Masroor Ahmad Paddar and colleagues reveal noncanonical roles of ATG5 and membrane ATG8ylation in regulating retromer assembly and function. They identify ATG5's unique non-autophagic role and show that CASM partially contributes to these phenotypes. Although the mechanism by which ATG8ylation regulates the retromer remains unclear, the findings provide **important** insights with **solid** supporting evidence.

*For correspondence: vderetic@salud.unm.edu

**Abstract** ATG5 is one of the core autophagy proteins with additional functions such as noncanonical membrane atg8ylation, which among a growing number of biological outputs includes control of tuberculosis in animal models. Here, we show that ATG5 associates with retromer's core components VPS26, VPS29, and VPS35 and modulates retromer function. Knockout of ATG5 blocked trafficking of a key glucose transporter sorted by the retromer, GLUT1, to the plasma membrane. Knockouts of other genes essential for membrane atg8ylation, of which ATG5 is a component, affected GLUT1 sorting, indicating that membrane atg8ylation as a process affects retromer function and endosomal sorting. The contribution of membrane atg8ylation to retromer function in GLUT1 sorting was independent of canonical autophagy. These findings expand the scope of membrane atg8ylation to specific sorting processes in the cell dependent on the retromer and its known interactors.

## Introduction

The canonical autophagy pathway, ubiquitous in eukaryotes, is manifested by the emergence of intracellular membranous organelles termed autophagosomes that capture cytoplasmic cargo destined for degradation in lysosomes (*Morishita and Mizushima, 2019*). Progress has been made in understanding molecular mechanisms governing canonical autophagosome biogenesis in mammalian cells (*Cook and Hurley, 2023*) including ATG9A vesicles (*Nguyen et al., 2023*; *Olivas et al., 2023*; *Kannangara et al., 2021*; *Ren et al., 2023*; *Broadbent et al., 2023*) and membranous prophagophores (*Kumar et al., 2021a*) as precursors to LC3-positive phagophores, their reshaping as cups (*Mohan et al., 2024*) and dynamic interactions with the omegasomes (*Axe et al., 2008*; *Nähse et al., 2023*),

phagophore expansion through lipid transfer (*Valverde et al., 2019*; *Maeda et al., 2019*; *Dabrowski et al., 2023*), cargo recognition and sequestration (*Turco et al., 2019*; *Lamark and Johansen, 2021*), closure of the phagophore to form a double membrane phagosomes (*Flower et al., 2020*; *Takahashi et al., 2019*; *Takahashi et al., 2018*; *Javed et al., 2023*), and fusion of autophagosomes with endosomal and lysosomal organelles leading to cargo degradation (*Zhao and Zhang, 2019*) or secretion (*Ponpuak et al., 2015*).

However, and contemporaneously with these major advances in understanding canonical autophagy, it has become evident that autophagy genes (ATGs) (*Yamamoto et al., 2023*) have additional functions in mammalian cells that do not fit the canonical model (*Galluzzi and Green, 2019*; *Deretic and Lazarou, 2022*). Among such noncanonical manifestations are LAP (LC3-associated phagocytosis) (*Sanjuan et al., 2007*), LANDO (LC3-associated endocytosis) (*Heckmann et al., 2019*), LAM (LC3-associated micropinocytosis) (*Sønder et al., 2021*), CASM (conjugation of ATG8 to single membranes) (*Goodwin et al., 2021*; *Hooper et al., 2022*), VAIL (v-ATPase-ATG16L1-induced LC3 lipidation) (*Xu et al., 2019*; *Fischer et al., 2020*; *Xu et al., 2022*), EVAC (ER-phagy mediated by the V-ATPase-ATG16L1–LC3C axis) (*Sun et al., 2023*), LyHYP (lysosomal hypersensitivity phenotype) (*Wang et al., 2023*), membrane damage repair (*Claude-Taupin et al., 2021*; *Jia et al., 2022*; *Kaur et al., 2023*; *Corkery et al., 2023*), and two distinct forms of secretory autophagy (*Ponpuak et al., 2015*), SALI (secretory autophagy during lysosome inhibition) (*Solvik et al., 2022*) and LDELS (LC3-dependent EV loading and secretion) (*Leidal et al., 2020*). These processes depend on or are associated with the phospholipid conjugation cascade of mammalian ATG8 proteins (mATG8s) (*Mizushima, 2020*) and collectively (including the canonical autophagy) represent diverse manifestations of membrane atg8ylation as a broad membrane stress, damage, and remodeling response (*Deretic and Lazarou, 2022*).

The factors governing membrane atg8ylation include two enzymatic cascades with the ATG12–ATG5 and mATG8–phosphatidylethanolamine (PE) covalent conjugates as their products. The ATG12–ATG5 conjugate (*Mizushima, 2020*) combines with ATG16L1 (*Rao et al., 2024*) or TECPR1 to form E3 ligases (*Kaur et al., 2023*; *Corkery et al., 2023*; *Boyle et al., 2023*) to direct mATG8–PE conjugation and atg8ylation of target membranes. All known E3 enzymes contain the ATG12–ATG5 conjugate (*Kaur et al., 2023*; *Corkery et al., 2023*; *Mizushima, 2020*; *Boyle et al., 2023*). To form this conjugate, the ubiquitin like molecule ATG12 is activated by ATP, and transferred via E1 (ATG7) and E2 (ATG10) to ATG5. In preparation for the next step, an ATG12–ATG5 containing E3 ligase activates its substrate mATG8–ATG3 to transfer ATG8 and form an amide bond with aminophospholipids (*Mizushima, 2020*). However, there are additional branches of these conjugation cascades, whereby ATG12 can make a noncanonical sidestep conjugate with ATG3 (ATG12–ATG3) (*Leidal et al., 2020*; *Radoshevich et al., 2010*) which is enhanced in the absence of ATG5 (*Wang et al., 2023*). Apart from its role in atg8ylation, ATG5 (*Wang et al., 2023*; *Castillo et al., 2012*) and potentially other ATG genes (*Hwang et al., 2012*; *Gu et al., 2019*; *Eren et al., 2020*) have autophagy-independent functions. Many of the indications that such functions exist come from in vivo studies (*Wang et al., 2020*; *Virgin and Levine, 2009*; *Deretic and Wang, 2023*). For example, in the case of murine models of *Mycobacterium tuberculosis* infection, atg8ylation machinery protects against tuberculosis pathogenesis but Atg5 knockout has a particularly strong phenotype exceeding other atg8ylation genes, sugestiong that ATG5 possesses atg8ylation (mATG8s lipid conjugation) independent functions (*Wang et al., 2023*; *Castillo et al., 2012*; *Watson et al., 2012*; *Kimmey et al., 2015*; *Golovkine et al., 2023*; *Kinsella et al., 2023*). As recently reported, one such function involves LyHYP contributing to enhanced exocytosis specifically in cells devoid of ATG5 but not of other ATG genes (*Wang et al., 2023*). These developments indicate that ATG5 plays noncanonical roles in autophagy-independent vesicular transport events and homeostasis of the endolysosomal system.

Within the endosomal network, the mammalian retromer complex (*Seaman, 2021*) is one of the several systems regulating vectoral transport of membranes and proteins. These systems include, among others, HOPS, CORVET, retriever, CCC/commander, and retromer (*Seaman, 2021*; *Burstein et al., 2005*; *Bonifacino and Hurley, 2008*; *Balderhaar and Ungermann, 2013*; *McNally et al., 2017*; *Mallam and Marcotte, 2017*; *Shvarev et al., 2022*; *Healy et al., 2023*) complexes that often contain and sometimes share VPS subunits. Retromer is a heterotrimer of VPS26A/B, VPS29, and VPS35 subunits (*Haft et al., 2000*), first identified in yeast (*Seaman et al., 1998*). Retromer sorts an array of endosomal cargo in cooperation with cognate sorting nexins (SNX) (*Gallon and Cullen, 2015*).

Specifically, this includes SNX-BAR proteins SNX1/2-SNX5/6 (*Rojas et al., 2008*; *Wassmer et al., 2007*), the mammalian paralogs of yeast Vps5p and Vps17p (*Seaman et al., 1998*) which can function autonomously as an ESCPE-I complex (*Simonetti et al., 2019*), SNX-PX protein SNX3 (*Harterink et al., 2011*), and the SNX-FERM protein SNX27 containing PX, PDZ, and FERM domains (*Gallon et al., 2014*) which sorts various transporter proteins and signaling receptors including the glucose transporter GLUT1 (SLC2A1) (*Steinberg et al., 2013*). Retromer, together with adaptors, contributes to the complex protein sorting within the endosomal system (*Tu and Seaman, 2021*; *Buser and Spang, 2023*; *Carosi et al., 2023*). Here, using unbiased proteomic approaches and follow-up mechanistic analyses, we report that retromer complex is among ATG5 interactors. We show that membrane atg8ylation, of which ATG5 is an essential component, affects retromer-dependent cargo sorting. This function is independent of the canonical autophagy pathway or a well-studied form of noncanonical membrane atg8ylation, CASM. Our findings offer a paradigm shift connecting ATG5 and membrane atg8ylation with the retromer system and its function in the endosomal cargo sorting, expanding the scope of atg8ylation machinery and its special component ATG5 beyond the current models.

## Results

### ATG5 associates with the retromer complex

The unique aspects of Atg5 (*Figure 1A*), outside of its role in canonical autophagy, have strong ex vivo and important in vivo phenotypes (*Deretic and Wang, 2023*). Inactivation of Atg5 in myeloid lineage renders mice excessively susceptible to acute infection with *M. tuberculosis* attributed to excessive inflammation (*Wang et al., 2023*; *Castillo et al., 2012*; *Watson et al., 2012*; *Kimmey et al., 2015*; *Golovkine et al., 2023*; *Kinsella et al., 2023*), which did not extend to other phases of infection as tested here in a murine model of latent tuberculosis (*Mccune and Tompsett, 1956*; *Mccune et al., 1956*; *Scanga et al., 1999*, *Figure 1—figure supplement 1A–D*). In a previous study, we could only partially explain the cellular parameters associated with acute infection, which included degranulation in neutrophils from *Atg5*^fl/fl *Lyz2*^Cre (also referred to in prior literature as Atg5^fl/fl LysM-Cre⁺) mice (*Wang et al., 2023*). Furthermore, markers of LyHYP (lysosome hypersensitivity phenotype) (*Wang et al., 2023*) in ATG5 knockout (KO) cells, such as galectin 3 (Gal3) response, could not be explained by the exocytic phenomena (*Wang et al., 2023*). We thus sought to uncover additional processes (*Figure 1A*) affected by ATG5 at the intracellular level. To include factors beyond the known processes, we analyzed proteomic data obtained with APEX2-ATG5 (*Wang et al., 2023*) and compared APEX2-ATG5^WT with APEX2-ATG5^K130R (an ATG5 mutant deficient in conjugation to ATG12) in cells treated with LLOMe, an agent routinely used to cause lysosomal damage (*Jia et al., 2020b*; *Aits et al., 2015*; *Thiele and Lipsky, 1990*; *Eriksson et al., 2020*; *Bonet-Ponce et al., 2020*; *Tan and Finkel, 2022*, *Figure 1B*, *Figure 1—figure supplement 2A, B*). The proximity biotinylation (APEX2)-based proteomic data indicated the presence in the vicinity of APEX2-ATG5 of several protein complexes that control important membrane trafficking pathways including retromer (*Figure 1B*). Beside the retromer VPS subunits, there were very few other VPS proteins (*Figure 1—figure supplement 2C*). VPS proteins were observed in ATG5 proteomic data by others (*Baines et al., 2022*, *Figure 1—figure supplement 2D*). VPS26 and VPS29 showed statistically significant increase in proximity to ATG5 in cells subjected to LLOMe treatment (*Figure 1—figure supplement 2A–C*). Using APEX2-SGALS1 proximity biotinylation LC–MS/MS as a control, we did not observe enrichment of retromer subunits in cells treated with LLOMe (*Figure 1—figure supplement 1* S2E).

Association of endogenous ATG5 with endogenous retromer subunits (VPS26A, VPS29 and VPS35) was tested in coimmunoprecipitation (co-IP) experiments (*Figure 1C–E*). The interaction between retromer and ATG5 was detected in these experiments and showed increased association in cells subjected to lysosomal damage with LLOMe (*Figure 1C–E*). Thus, ATG5, a key component of the known atg8ylation E3 ligases (*Mizushima, 2020*; *Deretic and Klionsky, 2024*), is found in protein complexes with the retromer and this association is enhanced upon lysosomal damage.

### Retromer affects a subset of responses to lysosomal damage

During lysosomal damage a number of processes are set in motion to repair, remove, and regenerate/replenish lysosomes (*Jia et al., 2022*; *Jia et al., 2020b*; *Jia et al., 2018*; *Nakamura et al., 2020*; *Jia et al., 2019*; *Eapen et al., 2021*) including membrane atg8ylation-dependent processes of repair

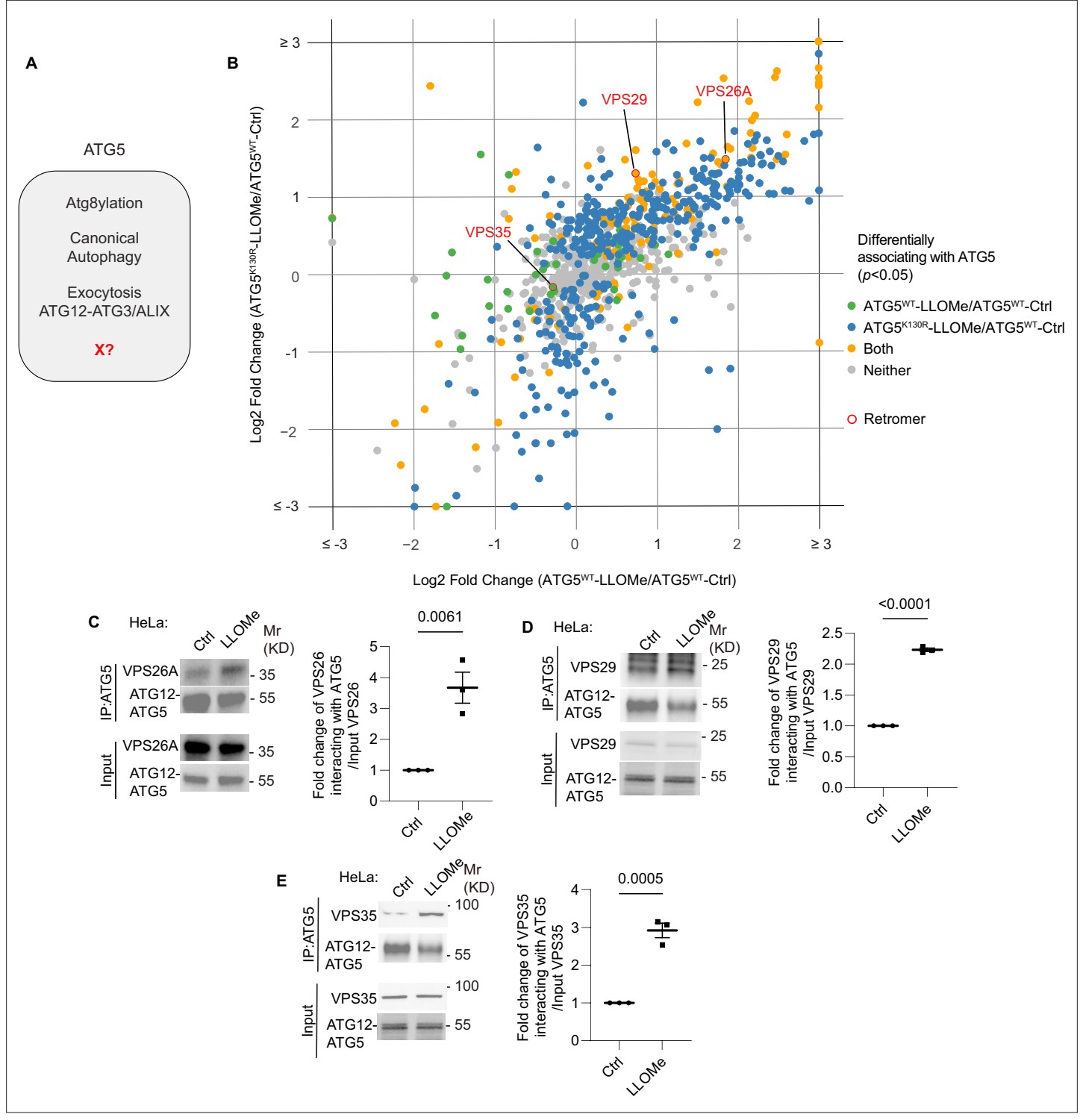

**Figure 1.** ATG5 interacts with retromer. (**A**) ATG5 functions. X, a postulated additional function. (**B**) 2D scatter plot (log2 fold changes; color coded p value cutoff matrix for comparisons between samples as per the lookup table) of proximity biotinylation LC/MS/MS datasets: FlpIn-HeLa[APEX2-ATG5-WT] cells (*X*-axis) and FlpIn-HeLa[APEX2-ATG5-K130R] (*Y*-axis) treated with 2 mM LLOMe for 30 min ratioed vs. HeLa[APEX2-ATG5-WT] without LLOMe treatment (Ctrl). Coimmunoprecipitation (co-IP) analyses and quantification of VPS26A (**C**), VPS29 (**D**), and VPS35 (**E**) interaction with ATG5 in HeLa cells treated with or without 2 mM LLOMe for 30 min. Data, means ± SE (*n* = 3); unpaired *t*-test; p values indicated above the bars.

The online version of this article includes the following source data and figure supplement(s) for figure 1:

**Source data 1.** PDF files containing original immunoblots for *Figure 1* indicating relevant bands.

*Figure 1 continued on next page*

*Figure 1 continued*

**Source data 2.** Original files for immunoblots displayed in *Figure 1*.

**Source data 3.** Numerical values for quantification in graphs.

**Figure supplement 1.** Cornell model of *M. tuberculosis* latent infection in mice and effects of Atg5 loss in myeloid lineage on disease reactivation.

**Figure supplement 2.** ATG5 interactome analysis.

by ESCRT machinery (*Corkery et al., 2024*) and by lipid transfer via ATG2 (*Tan and Finkel, 2022*; *Cross et al., 2023*). We have reported that ATG5 KO cells display elevated Gal3 response to lysosomal damage agents including LLOMe (*Wang et al., 2023*). Gal3 is one of the well-characterized sentinel galectins alerting cellular homeostatic systems to lysosomal damage (*Jia et al., 2022*; *Jia et al., 2020b*; *Aits et al., 2015*; *Jia et al., 2018*; *Jia et al., 2020a*). Based on the observed interactions between ATG5 and retromer, we tested whether retromer, like ATG5 (*Wang et al., 2023*), affected Gal3 response to lysosomal damage. We used the previously established quantitative high content microscopy (HCM) approach, which provides unbiased machine-driven image acquisition and data analysis (*Kumar et al., 2021a*; *Claude-Taupin et al., 2021*; *Jia et al., 2020b*; *Jia et al., 2018*; *Jia et al., 2020a*). In the experiments herein HCM was based on >500 valid primary objects/cells per well and a minimum of 5 wells per sample, with the independent biological replicates being $n \geq 3$ ($\geq 3$ separate plates) as described (*Wang et al., 2023*). Upon treatment with LLOMe, HeLa<sup>VPS35-KO</sup> cells had increased Gal3 puncta relative to HeLa<sup>WT</sup> cells, comparable to HeLa<sup>ATG5-KO</sup> (*Figure 2A, B*, *Wang et al., 2023*). The Gal3 phenotype in VPS35 knockout cells was partially complemented by expression of GFP-VPS35 (*Figure 2C, D*). The observed elevated Gal3 recruitment to damaged lysosomes in the absence of VPS35 cannot be explained by increased pools of LAMP2 organelles in the cells, since the overall complement of lysosomes (quantified by LAMP2 puncta/cell) did not increase in LLOMe treated HeLa cells (*Figure 2—figure supplement 1A, B*). Another marker of lysosomal damage, ubiquitination response (*Papadopoulos et al., 2017*), was elevated in both VPS35 and ATG5 deficient cells (*Figure 2E, F*). This phenotype was confirmed in another cell line, Huh7 (*Figure 2—figure supplement 1C, D*). Thus, VPS35 and ATG5 defects have a similar effect on lysosomes, previously characterized and termed lysosomal hypersensitivity phenotype/LyHYP (*Wang et al., 2023*).

Not all aspects of LyHYP, previously observed in ATG5 knockouts (*Wang et al., 2023*), were seen in VPS35 KO cells. In the case of ATG5 inactivation (Huh7<sup>ATG5-KO</sup> and HeLa<sup>ATG5-KO</sup>), the recruitment of ALIX, an ESCRT component involved in early stages of lysosomal repair (*Jia et al., 2020b*; *Skowyra et al., 2018*; *Radulovic et al., 2018*) is abrogated (*Wang et al., 2023*), whereas in VPS35 KO cells (Huh7<sup>VPS35-KO</sup> and HeLa<sup>VPS35-KO</sup>) ALIX was efficiently recruited to lysosomes upon LLOMe treatment (*Figure 2G, H*, *Figure 2—figure supplement 1E, F*).

The elevated Gal3 and ubiquitin markers of increased lysosomal damage in cells with inactivated VPS35 or ATG5 suggest that both may play a related role in lysosomal resilience to membrane damage, which complements the previously observed effects of aberrant VPS35/retromer on lysosomal morphology and proteolytic capacity (*Cui et al., 2019*; *Daly et al., 2023*). The dissimilarities between ATG5 and retromer effects on the ALIX recruitment component can be explained by the previously described perturbed function of the ATG conjugation machinery specifically in ATG5 KO cells affirming the unique additional roles of ATG5 (*Wang et al., 2023*). Nevertheless, the striking similarities between ATG5 KO and VPS35 KO effects on heightened Gal3 and ubiquitin responses to lysosomal membrane damage suggest that the two systems (membrane atg8ylation and retromer) closely intersect.

## ATG5 affects retromer function

To avoid the confounding, and possibly indirect, effects of lysosomal damage on endolysosomal sorting and trafficking in cells treated with LLOMe, it was necessary to test the effects of ATG5 KO on retromer function under basal conditions with unperturbed lysosomes. Our co-IP analyses with endogenous proteins (*Figure 1C–E*) indicated that ATG5 can be found in protein complexes with retromer subunits even without the lysosomal damage. We further confirmed this in co-IPs showing that GFP-VPS35 and YFP-VPS29 were found in complexes with endogenous ATG5 in resting cells, whereas an isotype IgG control did not pulldown retromer components (*Figure 3A*). Reverse co-IPs

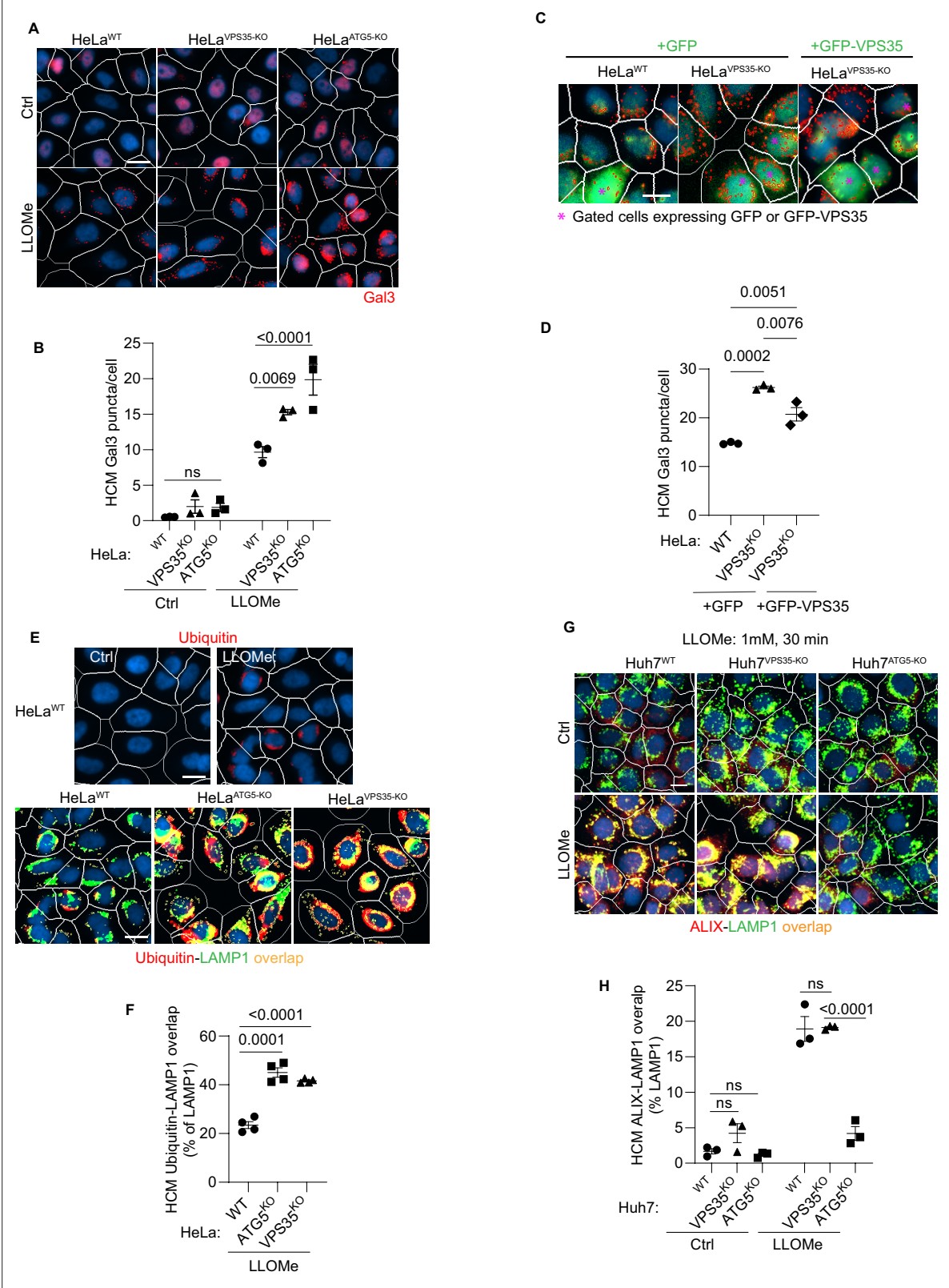

**Figure 2.** Retromer affects lysosomal sensitivity to damage. (**A**, **B**) High content microscopy (HCM) imaging and quantification of Gal3 response (puncta/cell of endogenous Gal3 profiles stained for immunofluorescence) in HeLa^WT, HeLa^ATG5-KO, and HeLa^VPS35-KO cells subjected to lysosomal damage by Leu-Leu-O-Me ester hydrobromide (LLOMe; 2 mM, 30 min). HCM, an unbiased machine-driven image acquisition and data analysis based on presets of >500 valid primary objects/cells per well (representative images shown; white mask, cell; red masks, Gal3 puncta), with a minimum of 5 wells per

*Figure 2 continued on next page*

*Figure 2 continued*

sample (sampling error), and $n \geq 3$, independent biological replicates (experimental error) in separate 96-well plates. Data, means ± SE ($n = 3$), two-way ANOVA with Tukey's multiple comparisons. (**C, D**) Complementation analysis of VPS35[KO] LyHYP (lysosome hypersensitivity) phenotype monitored by HCM quantification of Gal3 puncta in HeLa[VPS35-KO] cells transfected with GFP (control) or GFP-VPS35 expressing plasmids. Data, means ± SE ($n = 3$); one-way ANOVA with Tukey's multiple comparisons. (**E, F**) Comparative HCM analysis of ubiquitin (immunofluorescence; FK2 antibody) response to lysosomal damage (LLOMe; 2 mM, 30 min) in HeLa[ATG5-KO] and HeLa[VPS35-KO] cells. Yellow profiles, colocalization of ubiquitin and LAMP1. Top panels, ubiquitin immunostaining alone. Data, means ± SE ($n = 4$), two-way ANOVA with Tukey's multiple comparisons. (**G, H**) HCM quantification of ALIX localization to endolysosomal compartments (% of LAMP1 profiles positive for ALIX immunostaining) in Huh7[WT], Huh7[ATG5-KO], and Huh7[VPS35-KO] cells following lysosomal damage. Yellow profiles, colocalization of ALIX and LAMP1. Data, means ± SE ($n = 3$); two-way ANOVA with Tukey's multiple comparisons. HCM images in all relevant panels, examples from a bank of unbiased operator-independent machine-collected and algorithm-processed fields containing a minimum of 500 primary objects/cells per well (5 wells minimum per 96-well plate; 3 plates minimum), per cell line/condition.

The online version of this article includes the following source data and figure supplement(s) for figure 2:

**Source data 1.** Numerical values for quantification in graphs.

**Figure supplement 1.** Retromer affects a subset of responses to lysosomal damage.

**Figure supplement 1—source data 1.** Numerical values for quantification in graph.

with GFP-VPS35 and YFP-VPS29 detected endogenous ATG5 and ATG12–ATG5 complexes in GFP/YFP immunoprecipitates (*Figure 3B*).

Having established that ATG5 and retromer interact under basal conditions, we tested the effects of ATG5 KO on localization of a well-established cargo for retromer-dependent sorting, GLUT1 (*Steinberg et al., 2013*; *McGough et al., 2014*). We used Huh7 cells, where we generated a panel of membrane atg8ylation and VPS35 CRISPR knockouts (*Figure 3—figure supplement 1A*), because they have a well-defined localization of GLUT1 at the plasma membrane or as intracellular profiles (*Figure 3C*), fully amenable (and comparatively more reliable than in HeLa cells) to unbiased quantification by HCM. ATG5 KO in Huh7 cells resulted in redistribution of GLUT1 from the plasma membrane to intracellular punctate profiles (*Figure 3D*, *Figure 3—figure supplement 1B*). This phenotype was identical to the one observed in VPS35 KO (*Figure 3D*, *Figure 3—figure supplement 1B*). The increased intracellular GLUT1 puncta overlapped with the lysosomal marker LAMP2 in both ATG5 KO and VPS35 KO cells (*Figure 3C, E*, *Figure 3—figure supplement 1C*). A similar albeit morphologically less distinct phenotype was observed in HeLa cells (*Figure 3—figure supplement 1D–H*). GLUT1 is sorted by retromer via VPS26's interactor SNX27, with GLUT1 (and other similar cargo) being captured by the PDZ domain of SNX27 (*Gallon et al., 2014*; *Steinberg et al., 2013*). In WT cells, SNX27 was found in abundant intracellular profiles (*Figure 3F*). In both ATG5 KO and VPS35 KO cells, GLUT1 and SNX27 relocalized to the same intracellular compartment (*Figure 3F, G*; *Figure 3—figure supplement 1I, J*). We carried out lysosomal purification using the well-established LysoIP method (*Jia et al., 2020b*; *Eapen et al., 2021*; *Jia et al., 2020a*; *Wyant et al., 2018*), and quantified levels of GLUT1 and SNX27 in lysosomal preparations that were positive for LAMP2 and devoid of GM130 (Golgi) and PDI (ER) (*Figure 3H, I*). Both GLUT1 and SNX27 were enriched in lysosomal preparations from ATG5 KO and VPS35 KO cells relative to the WT cells (*Figure 3H, I*). These experiments biochemically identify the lysosomes as a compartment to which GLUT1 and SNX27 relocalize in ATG5 KO and VSP35 KO cells. The above findings show that a loss of ATG5 affects retromer-controlled trafficking events.

## Membrane atg8ylation apparatus contributes to retromer function in GLUT1 sorting

ATG5 is a part of the membrane atg8ylation machinery (*Deretic and Lazarou, 2022*; *Mizushima, 2020*; *Deretic and Klionsky, 2024*), also known under the term 'LC3 lipidation' (*Kabeya et al., 2000*), which functions within the canonical autophagy pathway but is also engaged in a wide array of noncanonical processes (*Deretic and Lazarou, 2022*; *Kaur et al., 2023*; *Corkery et al., 2023*; *Boyle et al., 2023*; *Durgan and Florey, 2022*; *Deretic et al., 2024*). We tested whether other known components of the atg8ylation apparatus affect retromer. The KOs in Huh7 cells of the components of the prototypical atg8ylation E3 ligase (*Deretic and Lazarou, 2022*; *Mizushima, 2020*) included: ATG5 and ATG16L1, as well as the previously characterized Huh7 lines knocked out for E1 and E2 enzymes, ATG7 (*Wang et al., 2023*) and ATG3 (*Jia et al., 2022*). These KOs all caused entrapment of GLUT1 in intracellular punctate profiles (*Figure 4A, B*) as well as an increase in GLUT1 colocalization with lysosomes (GLUT1[+] LAMP2[+] profiles; *Figure 4C, D*). We tested whether retromer component levels

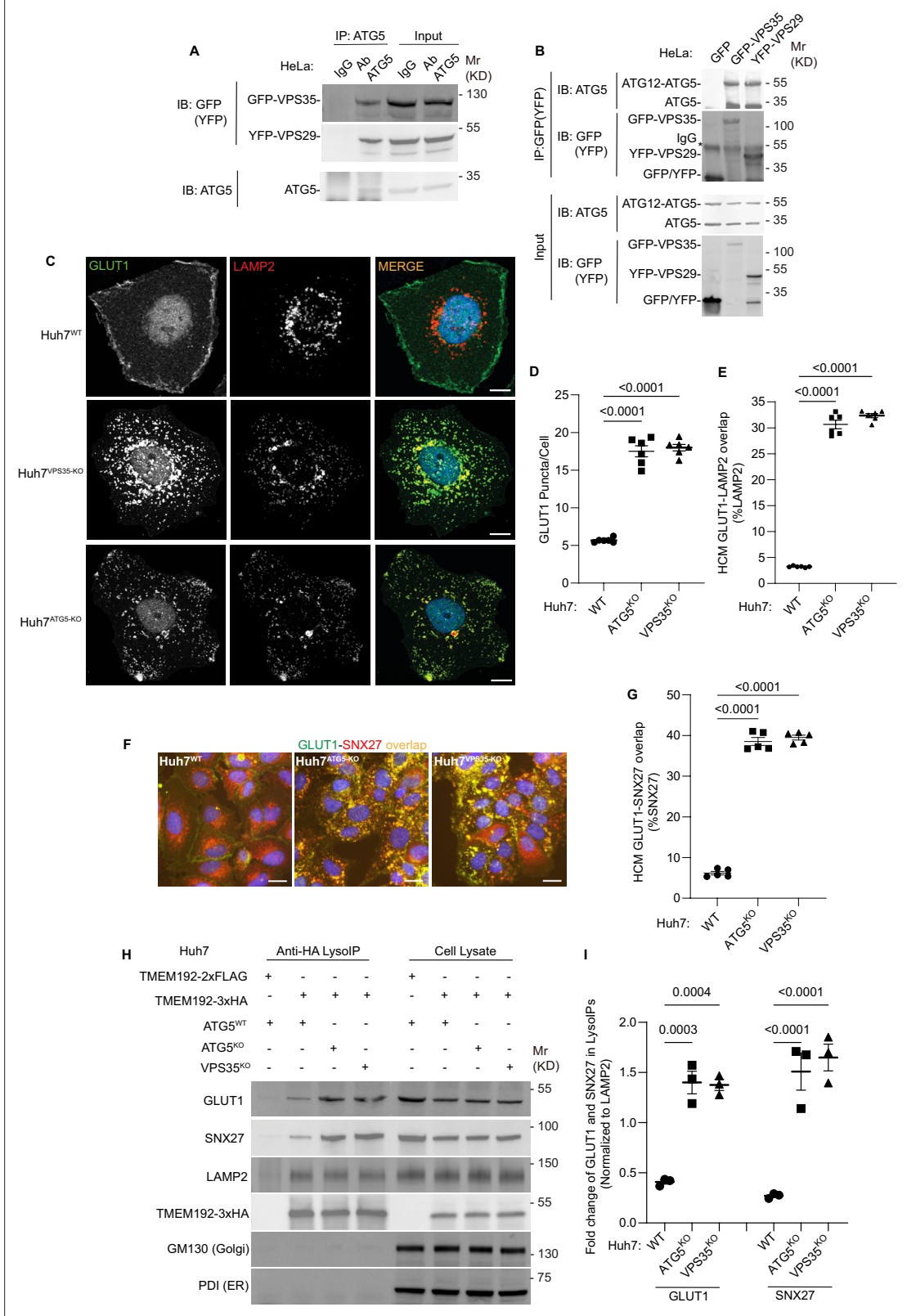

**Figure 3.** ATG5 affects sorting of the retromer cargo GLUT1. (**A**) Coimmunoprecipitation (co-IP) analysis of endogenous ATG5 with GFP-VPS35 or YFP-VPS29 (transient transfection). (**B**) Reverse co-IP analysis GFP-VPS35 or YFP-VPS29 (transient transfection) and endogenous ATG5. Note that different antibodies were used in panels A and B: ab108327 (Abcam) recognizing preferentially the conjugated form of ATG5 and abASA-B0113 (Novateinbio), recognizing both conjugated and unconjugated ATG5. (**C**) Confocal images illustrating localization of GLUT1 and LAMP2 in Huh7^WT, Huh7^VPS35-KO, and

*Figure 3 continued on next page*

*Figure 3 continued*

Huh7[ATG5-KO] cells. Scale bar, 10 μm. High content microscopy (HCM) quantification of GLUT1 (endogenous protein immunostaining) puncta/cell (**D**) and GLUT1 colocalization with LAMP2 (% of LAMP2 profiles positive for GLUT1; overlap area) (**E**) in Huh7[WT], Huh7[ATG5-KO], and Huh7[VPS35-KO] cells. Scale bar, 20 μm. Data, means ± SE (*n* = 6); one-way ANOVA with Tukey's multiple comparisons. (**F, G**) HCM quantification of GLUT1-SNX27 overlap (% of SNX7 area positive for GLUT1) in Huh7[WT], Huh7[ATG5-KO], and Huh7[VPS35-KO] cells. Scale bar, 20 μm. Data, means ± SE (*n* = 5); one-way ANOVA with Tukey's multiple comparisons. (**H, I**) Immunoblot analysis and quantification of proteins in lysosomes purified/enriched by LysoIP (immunoisolation with TMEM192-3xHA) from Huh7[WT], Huh7[ATG5-KO], and Huh7[VPS35-KO] cells. TMEM192-2xFLAG, negative control. Data, means ± SE (*n* = 3), one-way ANOVA with Tukey's multiple comparisons. HCM images in panel F, examples from a bank of unbiased operator-independent machine-collected and algorithm-processed fields containing a minimum of 500 primary objects/cells per well (5 wells minimum per 96-well plate; 3 plates minimum), per cell line.

The online version of this article includes the following source data and figure supplement(s) for figure 3:

**Source data 1.** PDF files containing original immunoblots for *Figure 3* indicating relevant bands.

**Source data 2.** Original files for immunoblots displayed in *Figure 3*.

**Source data 3.** Numerical values for quantification in graphs.

**Figure supplement 1.** Membrane atg8ylation regulates retromer function.

**Figure supplement 1—source data 1.** Numerical values for quantification in graphs.

were altered in atg8ylation mutants and found no change in total cellular levels of VPS26, VPS29, and VPS35, tested in ATG3, ATG5, and ATG7 KO Huh7 cells (*Figure 4—figure supplement 1A, B*) as well as in ATG5 KO HeLa cells (*Figure 4—figure supplement 1C, D*). Thus, the entire ATG16L1-dependent atg8ylation apparatus is required to maintain proper retromer-dependent sorting.

Furthermore, comparison of the previously characterized HeLa[Hexa-KO] cells (*Nguyen et al., 2016*), with all principal mATG8s inactivated, with their isogenic parental HeLa[WT] cells, showed increased GLUT1 on lysosomes in HeLa[Hexa-KO] (GLUT1[+] LAMP2[+] profiles; *Figure 4E, F*). We observed a similar phenotype in separate triple knockouts (TKOs) of LC3 subfamily and GABARAP subfamily of mATG8s (*Nguyen et al., 2016*) compared to HeLa[Hexa-KO] (*Figure 4—figure supplement 1E*). Thus, at least two or more mATG8s from two different mATG8 subclasses (LC3s and GABARAPs) and the whole membrane atg8ylation machinery were engaged in and required for proper GLUT-1 sorting.

## Canonical autophagy cannot explain effects of membrane atg8ylation on GLUT1 sorting

Membrane atg8ylation and mATGs play roles in diverse processes (*Deretic and Lazarou, 2022*) including canonical autophagy (*Morishita and Mizushima, 2019*). We tested whether the participation of atg8ylation in canonical autophagy, previously reported to affect retromer-dependent cargo sorting under nutrient-limiting conditions (*Roy et al., 2017*; *Carosi et al., 2024*), is the reason for reduced trafficking of GLUT1 in our experiments. We used KOs in Huh7 cells of ATG13 and FIP200/RB1CC1, two obligatory components of the canonical autophagy initiation machinery (*Morishita and Mizushima, 2019*), to test whether canonical autophagy under the basal conditions used in our study was responsible for the effects on GLUT1 sorting. We found that in the Huh7 cells knocked out for ATG13 (*Wang et al., 2023*) or FIP200 (*Jia et al., 2022*), there was no increase in intracellular GLUT1 puncta in contrast to the increase observed with the isogenic ATG5 and VPS35 mutants (*Figure 5A, B*). This was mirrored by no increase of GLUT1[+]LAMP2[+] profiles in ATG13 and FIP200 KO cells whereas colocalization between GLUT1 and LAMP2A increased in ATG5 and VPS35 mutants (*Figure 5C, D*). To further confirm that autophagy initiation mutant cells retained the ability to perturb GLUT1 trafficking due to defective atg8ylation, we knocked down ATG5 in FIP200 KO cells (*Figure 4—figure supplement 1F*) and found that GLUT1 puncta and GLUT1[+]LAMP2[+] profiles increased even in the FIP200 KO background with the effects nearing those of VPS35 knockout (*Figure 5E–G*, *Figure 4—figure supplement 1G*), with the difference between VPS35 KO and ATG5 KD being attributable to residual ATG5 levels in cells subjected to siRNA knockdowns. Thus, we conclude that atg8ylation but not canonical autophagy affects retromer function under basal conditions and that a functional atg8ylation apparatus is required for proper sorting of the retromer cargo GLUT1.

## ATG5 and membrane atg8ylation machinery affect Rab7 localization

Rab7 is considered to be an interactor of the retromer (*Harrison et al., 2014*). Rab7 translocates in cells lacking the retromer component VPS35 to lysosomes (*Jimenez-Orgaz et al., 2018*) including

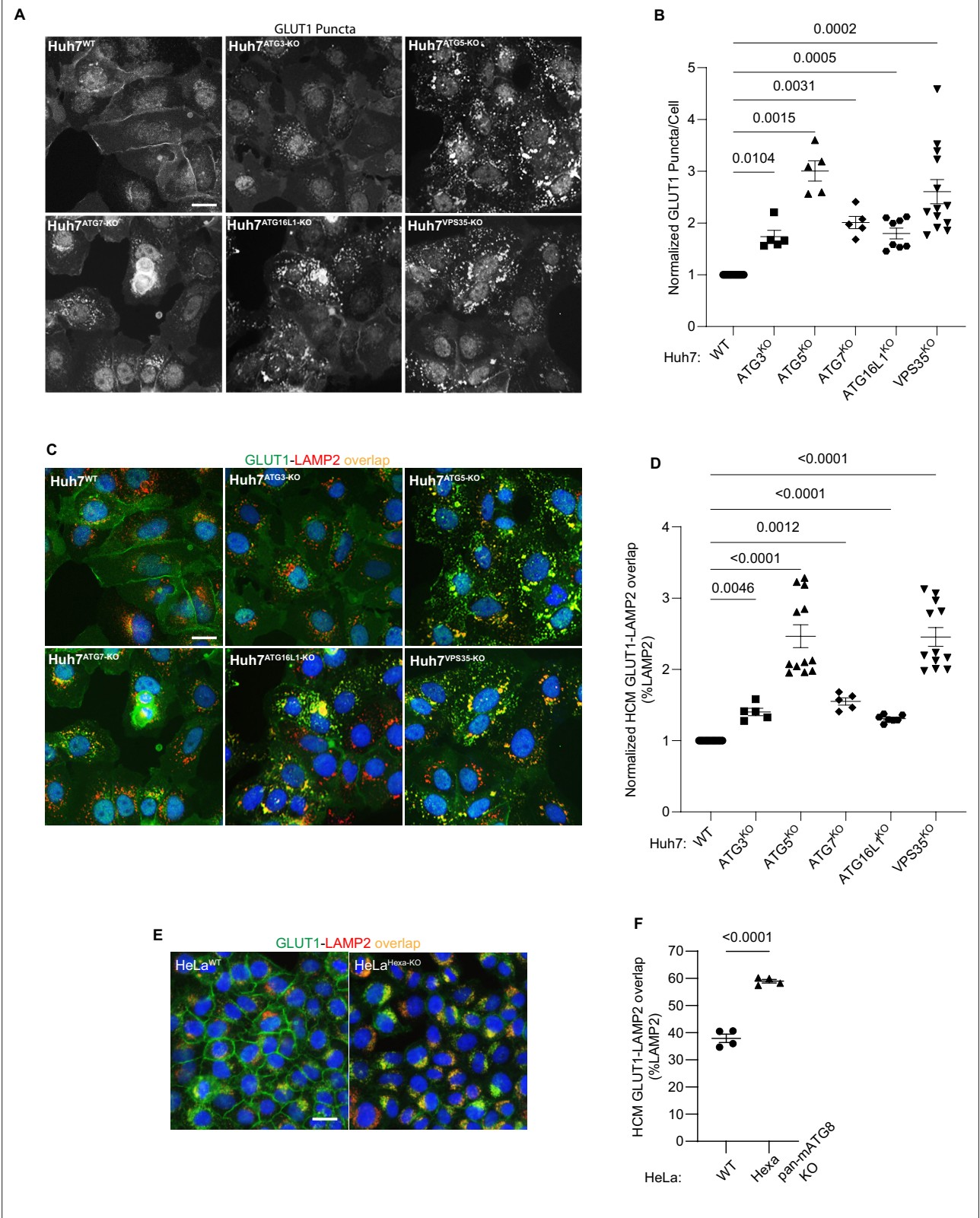

**Figure 4.** Membrane atg8ylation apparatus affects sorting of the retromer cargo GLUT1. (**A, B**) High content microscopy (HCM) quantification of GLUT1 (endogenous protein immunostaining) puncta/cell in Huh7$^{WT}$, Huh7$^{ATG3-KO}$, Huh7$^{ATG5-KO}$, Huh7$^{ATG7-KO}$, Huh7$^{ATG16L1-KO}$, and Huh7$^{VPS35-KO}$ cells. Scale bar, 20 μm. Data, means ± SE ($n = 5$); one-way ANOVA with Tukey's multiple comparisons. (**C, D**) HCM quantification of GLUT1 (endogenous protein immunostaining) colocalization with LAMP2 (% of LAMP2 profiles positive for GLUT1; overlap area) in Huh7$^{WT}$, Huh7$^{ATG3-KO}$, Huh7$^{ATG5-KO}$, Huh7$^{ATG7-KO}$,

*Figure 4 continued on next page*

*Figure 4 continued*

Huh7[ATG16L1-KO], and Huh7[VPS35-KO] cells. Scale bar, 20 μm. Data, means ± SE (*n* = 5); one-way ANOVA with Tukey's multiple comparisons. (**E, F**) HCM quantification of GLUT1 (endogenous protein immunostaining) colocalization with LAMP2 (% of LAMP2 profiles positive for GLUT1; overlap area) in HeLa[WT], and HeLa[Hexa-KO] cells. Scale bar, 20 μm. Data, means ± SE (*n* = 5); one-way ANOVA with Tukey's multiple comparisons. HCM images in all relevant panels, examples from a bank of unbiased operator-independent machine-collected and algorithm-processed fields containing a minimum of 500 primary objects/cells per well (5 wells minimum per 96-well plate; 3 plates minimum), per cell line.

The online version of this article includes the following source data and figure supplement(s) for figure 4:

**Source data 1.** Numerical values for quantification in graphs.

**Figure supplement 1.** ATG5 knockout has no effect on protein levels of retromer subunits.

**Figure supplement 1—source data 1.** PDF files containing original immunoblots for *Figure 4—figure supplement 1* indicating relevant bands.

**Figure supplement 1—source data 2.** Original files for immunoblots displayed in *Figure 4—figure supplement 1*.

**Figure supplement 1—source data 3.** Numerical values for quantification in graphs.

endolysosomal domains with lysosomaly positioned mTORC1 regulatory machinery (*Kvainickas et al., 2019*). We thus tested whether absence of ATG5 and components of the atg8ylation apparatus affected Rab7 localization. ATG5, VPS35 as well as knockouts of all major components of the atg8ylation in Huh7 cells, displayed increased Rab7[+] cytoplasmic profiles (*Figure 6A, B*) and overlap between Rab7 and lysosomes (LAMP2) (*Figure 6C*, *Figure 6—figure supplement 1A*). This was confirmed biochemically using lysosomal purification by LysoIP and immunoblotting for Rab7: there was an increase of Rab7 in lysosomal preparations from ATG5 and VPS35 KO cells (*Figure 6D, E*).

This effect extended to other components of the atg8ylation machinery, as KOs in ATG3, ATG7, and ATG16L1 caused a similar effect (*Figure 6A–C*, *Figure 6—figure supplement 1A*). In contrast, KOs in genes specific for canonical autophagy (ATG13 and FIP200) did not alter intracellular distribution of Rab7 (*Figure 6F–H*, *Figure 6—figure supplement 1B*). Rab7 distribution was affected by ATG5 knockdown in FIP200 KO cells (*Figure 6—figure supplement 1C–F*). These experiments mirror the effects of ATG5 and atg8ylation on GLUT1 trafficking, showing that ATG5 and atg8ylation machinery but not canonical autophagy is required for proper localization of Rab7.

## Agonists of endolysosomal atg8ylation process CASM affect GLUT1 sorting

Membrane atg8ylation has multiple presentations that include not only canonical autophagy but also encompass processes such as CASM (*Durgan and Florey, 2022*). We thus asked whether induction of CASM (*Cross et al., 2023*) could affect GLUT1 trafficking. Using LLOMe as one of the inducers of CASM (*Cross et al., 2023*), we detect increased membrane atg8ylation (LC3 puncta formation) in response to lysosomal damage (*Figure 7A, B*). When cells were treated with LLOMe (*Cross et al., 2023*) and another CASM inducer, monensin (*Goodwin et al., 2021*; *Florey et al., 2015*; *Fletcher et al., 2018*), GLUT1 trafficking to the plasma membrane was negatively affected and instead GLUT1 accumulated intracellularly and trafficked to lysosomes (*Figure 7C–E*, *Figure 7—figure supplement 1A*). Thus, lysosomal perturbation and induction of CASM may affect retromer-dependent sorting of GLUT1.

It was previously reported that under glucose starvation conditions, LC3A recruits TBC1D5, a GAP for Rab7 (*Popovic et al., 2012*; *Borchers et al., 2021*; *Stroupe, 2018*), away from this small GTPase (*Roy et al., 2017*) and a similar phenomenon could be occurring during CASM. However, under CASM-inducing conditions, no changes were detected (*Figure 7—figure supplement 1B–D*) in interactions between TBC1D5 and LC3A or in levels of VPS35 in LC3A co-IPs, a proxy for LC3A-TBC1D5-VPS29/retromer association. This suggests that CASM-inducing treatments and additionally bafilomycin A1 do not affect the status of the TBC1D5-Rab7 system. Moreover, expression of constitutively active Rab7[Q67L] did not promote trafficking of GLUT1 to plasma membrane but rather caused accumulation of intracellular GLUT1 puncta (*Figure 7F–H*).

## Membrane atg8ylation and endolysosomal homeostasis affect GLUT1 sorting

A complication of interpreting CASM as a membrane atg8ylation process responsible for effects on retromer-dependent sorting is that CASM is elicited by lysosomal perturbations using lysosomal pH

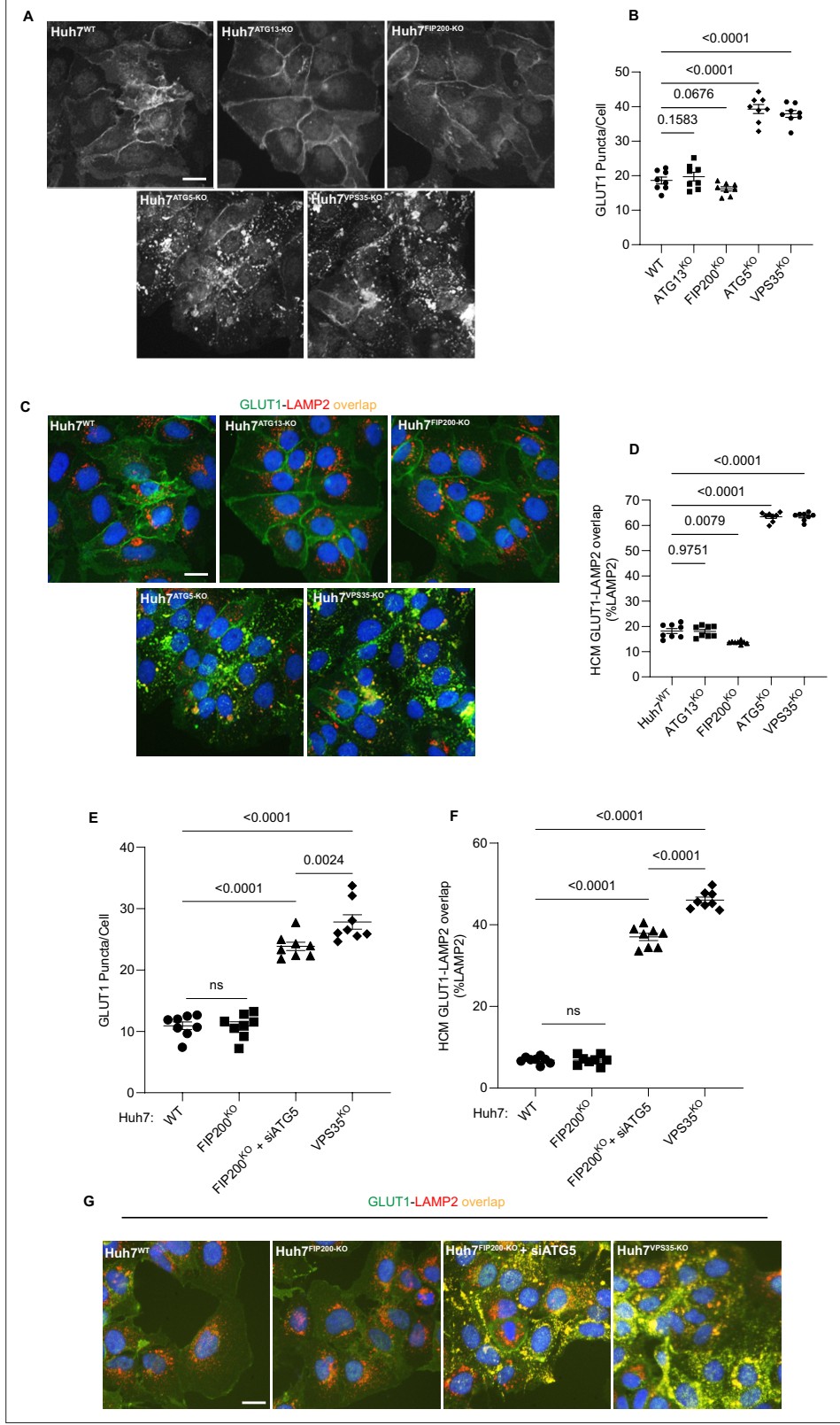

**Figure 5.** Canonical autophagy does not affect sorting of the retromer cargo GLUT1. (**A, B**) High content microscopy (HCM) quantification of GLUT1 (immunostaining of endogenous protein) puncta/cell in in Huh7[WT], Huh7[ATG13-KO], Huh7[FIP200-KO], Huh7[ATG5-KO], and Huh7[VPS35-KO] cells. Scale bar, 20 μm. Data, means ± SE ($n$ = 5); one-way ANOVA with Tukey's multiple comparisons. (**C, D**) HCM quantification of GLUT1 (immunostaining of endogenous

*Figure 5 continued on next page*

*Figure 5 continued*

protein) colocalization with LAMP2 (% of LAMP2 profiles positive for GLUT1; overlap area) in in Huh7[WT], Huh7[ATG13-KO], Huh7[FIP200-KO], Huh7[ATG5-KO], and Huh7[VPS35-KO] cells. Scale bar, 20 μm. Data, means ± SE (*n* = 5); one-way ANOVA with Tukey's multiple comparisons. (**E–G**) HCM quantification of GLUT1 (immunostaining of endogenous protein) puncta/cell (**E**) and GLUT1 colocalization with LAMP2 (% of LAMP2 profiles positive for GLUT1; overlap area) (**F**) in Huh7[WT], Huh7[FIP200-KO], Huh7[FIP200-KO + siATG5], and Huh7[VPS35-KO] cells. Scale bar, 20 μm. Data, means ± SE (*n* = 6); one-way ANOVA with Tukey's multiple comparisons. HCM images in all relevant panels, examples from a bank of unbiased operator-independent machine-collected and algorithm-processed fields containing a minimum of 500 primary objects/cells per well (5 wells minimum per 96-well plate; 3 plates minimum), per cell line.

The online version of this article includes the following source data for figure 5:

**Source data 1.** Numerical values for quantification in graphs.

modifying agents (*Goodwin et al., 2021*; *Durgan and Florey, 2022*; *Florey et al., 2015*; *Fletcher et al., 2018*), and lysosomal stress could be an independent factor preventing proper sorting of GLUT1. Endolysosomal membrane damage (LLOMe) or perturbations of luminal pH (monensin, bafilomycin A1) negatively affected GLUT1 sorting and caused it to accumulate in lysosomes (*Figure 7C–E*). Since bafilomycin A1 does not induce CASM (*Cross et al., 2023*) but disturbs luminal pH, we conclude that it is the less acidic luminal pH of the endolysosomal organelles, and not CASM, that is responsible for interference with the proper sorting of GLUT1. To additionally test this, we compared ATG16L1 full length (ATG16L1[FL]) and ATG16L1[E230] (*Rai et al., 2019*) for complementation of the GLUT1 sorting defect in ATG16L1 KO cells (*Figure 7I, J*). ATG16L1[E230] (*Rai et al., 2019*) lacks the key domain to carry out CASM via binding to V-ATPase (*Goodwin et al., 2021*; *Hooper et al., 2022*; *Xu et al., 2019*; *Fischer et al., 2020*; *Xu et al., 2022*) but retains capacity to carry out atg8ylation. Both ATG16L1[FL] and ATG16L1[E230] complemented mis-sorting of GLUT1 (*Figure 7I, J*). Collectively, these data indicate that it is not specifically an absence of CASM/VAIL but the absence of membrane atg8ylation in general that promotes GLUT1 mis-sorting.

The above experiments suggest the role of acidification and integrity of endolysosomal compartments in GLUT1 sorting. Membrane atg8ylation is important for lysosomal repair and homeostasis with specific downstream mechanisms. These processes include: (1) ATG2A, which transfers lipids during lysosomal repair (*Tan and Finkel, 2022*) and is engaged on damaged lysosomes upon membrane atg8ylation by LC3A (*Cross et al., 2023*), and (2) ESCRT-based lysosomal membrane repair (*Jia et al., 2020b*; *Skowyra et al., 2018*; *Radulovic et al., 2018*; *Zhen et al., 2021*), with membrane atg8ylation specifically recruiting an ESCRT regulator ALG2 (*Corkery et al., 2024*) and ESCRT-I component VPS37A (*Javed et al., 2023*). Hence, we tested whether these established systems, involved in maintenance of lysosomal membrane integrity, represented by ATG2 and VPS37, can explain the mechanism underlying the observed effects of membrane atg8ylation on retromer-dependent trafficking. When U2OS ATG2A/ATG2B double KO cells were compared to parental U2OS WT cells, we observed aberrant GLUT1 trafficking, presented as accumulation of cytoplasmic GLUT1 puncta and increased colocalization of GLUT1 with lysosomes (LAMP2A) (*Figure 8A–C*, *Figure 8—figure supplement 1A*). A similar effect was observed when the previously characterized VPS37A knockout cells (*Javed et al., 2023*) were tested (*Figure 8D–F*, *Figure 8—figure supplement 1B*). These effects were detected with or without induced lysosomal damage (*Figure 8A–F*).

We interpret these data as evidence that endolysosomal homeostasis and functionality, which is at all times maintained by membrane atg8ylation, even under basal conditions, contributes to proper function of retromer-dependent sorting of its cognate cargo such as GLUT1 (*Figure 8G*).

## Effect of ATG5 knockout on MPR sorting

We tested whether ATG5 affects cation-independent mannose 6-phosphate receptor (CI-MPR). For this, we employed the previously developed methods (*Figure 8—figure supplement 2A*) of monitoring retrograde trafficking of CI-MPR from the plasma membrane to the TGN (*Simonetti et al., 2019*; *Calcagni' et al., 2023*; *Seaman, 2004*; *Kvainickas et al., 2017*; *Zhang et al., 2023*). In the majority of such studies, CI-MPR antibody is allowed to bind to the extracellular domain of CI-MPR at the plasma membrane and its localization dynamics following endocytosis serves as a proxy for trafficking of CI-MPR. We used ATG5 KOs in HeLa and Huh7 cells and quantified by HCM retrograde trafficking to TGN of antibody-labeled CI-MPR at the cell surface, after being taken up by endocytosis

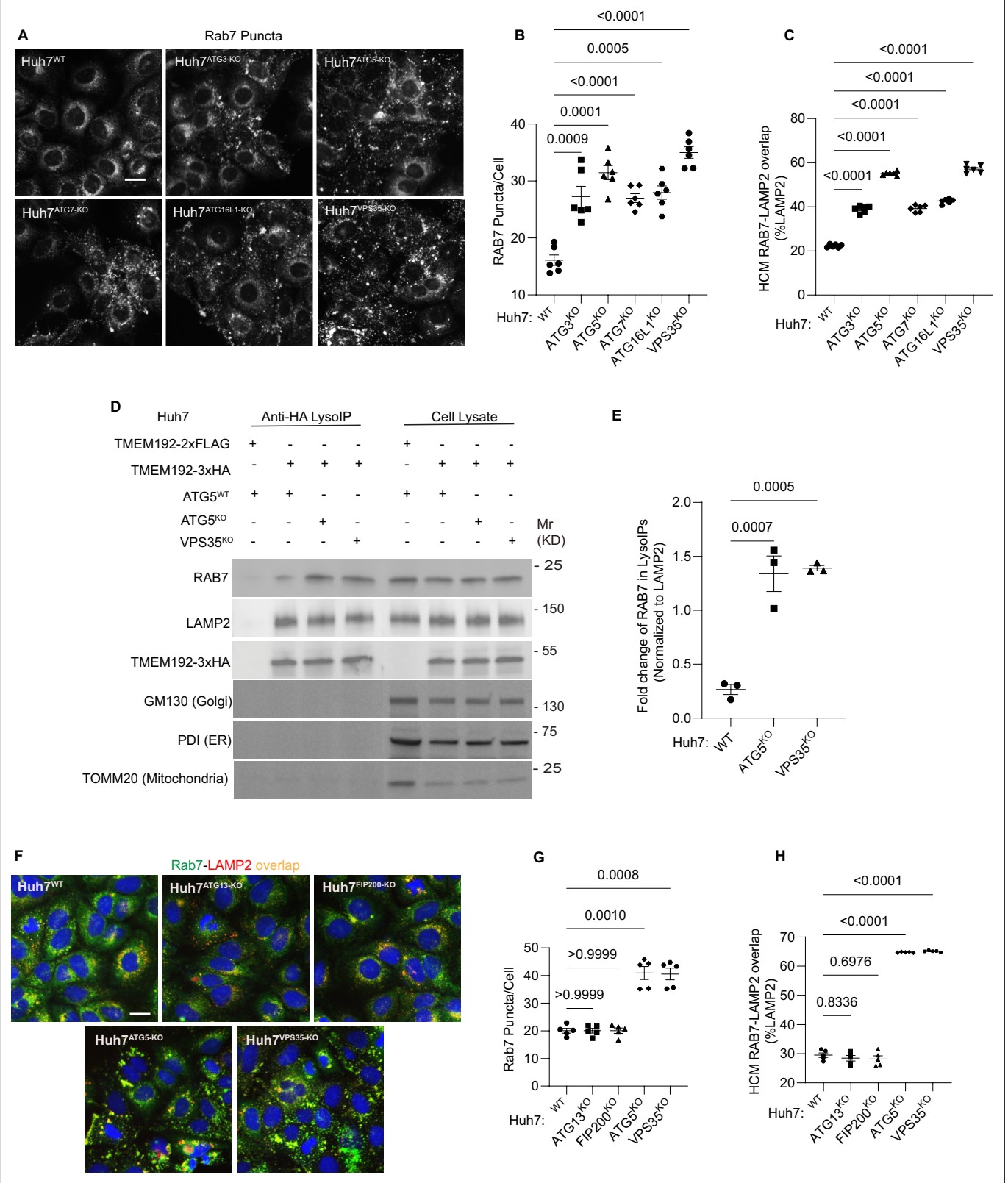

**Figure 6.** Membrane atg8ylation machinery is required for proper RAB7 localization. (**A–C**) High content microscopy (HCM) quantification of Rab7 (endogenous protein immunostaining) puncta/cell (**B**) and Rab7 colocalization with LAMP2 (% of LAMP2 profiles positive for GLUT1; overlap area) (**C**) in Huh7$^{WT}$, Huh7$^{ATG3-KO}$, Huh7$^{ATG5-KO}$, Huh7$^{ATG7-KO}$, Huh7$^{ATG16L1-KO}$, and Huh7$^{VPS35-KO}$ cells. Scale bar, 20 μm. Data, means ± SE (*n* = 6); one-way ANOVA with Tukey's multiple comparisons. (**D, E**) Immunoblot analysis and quantification of proteins in lysosomes purified/enriched by LysoIP (immunoisolation

*Figure 6 continued on next page*

*Figure 6 continued*

with TMEM192-3xHA) from Huh7[WT], Huh7[ATG5-KO], and Huh7[VPS35-KO] cells. TMEM192-2xFLAG, negative control. Data, means ± SE (*n* = 3), one-way ANOVA with Tukey's multiple comparisons. (**F–H**) HCM quantification of Rab7 (endogenous protein immunostaining) puncta/cell (**G**) and Rab7 colocalization with LAMP2 (% of LAMP2 profiles positive for GLUT1; overlap area) (**H**) in Huh7[WT], Huh7[ATG13-KO], Huh7[FIP200-KO], Huh7[ATG5-KO], and Huh7[VPS35-KO] cells. Scale bar, 20 μm. Data, means ± SE (*n* = 6); one-way ANOVA with Tukey's multiple comparisons. HCM images in all relevant panels, examples from a bank of unbiased operator-independent machine-collected and algorithm-processed fields containing a minimum of 500 primary objects/cells per well (5 wells minimum per 96-well plate; 3 plates minimum), per cell line.

The online version of this article includes the following source data and figure supplement(s) for figure 6:

**Source data 1.** PDF files containing original immunoblots for *Figure 6* indicating relevant bands.

**Source data 2.** Original files for immunoblots displayed in *Figure 6*.

**Source data 3.** Numerical values for quantification in graphs.

**Figure supplement 1.** Loss of membrane atg8ylation but not of canonical autophagy diverts of RAB7 to lysosomal compartments.

**Figure supplement 1—source data 1.** Numerical values for quantification in graphs.

and allowed to undergo intracellular sorting, followed by fixation and staining with TGN46 antibody. There was a minor but statistically significant reduction in CI-MPR overlap with TGN46 in HeLa[ATG5-KO] that was comparable to the reduction in HeLa cells when VPS35 was depleted by CRISPR (HeLa[VPS35-KO]) (*Figure 8—figure supplement 2B, C*). Morphologically, endocytosed Ab-CI-MPR appeared dispersed in both HeLa[ATG5-KO] and HeLa[VPS35-KO] cells relative to HeLa[WT] cells (*Figure 8—figure supplement 2D*). Similar HCM results were obtained with Huh7 cells (WT vs. ATG5KO; *Figure 8—figure supplement 2E, F*). We interpret these data as evidence of indirect action of ATG5 KO on CI-MPR sorting via membrane homeostasis, although we cannot exclude a direct sorting role via retromer. We favor the former interpretation based on the magnitude of the effect and the controversial nature of retromer engagement in sorting of CI-MPR (*Seaman, 2021*; *Simonetti et al., 2019*; *Buser and Spang, 2023*; *Cui et al., 2019*; *Kvainickas et al., 2017*).

## ATG5 effects on retromer function exceed those of membrane atg8ylation

Finally, we addressed the strength of effects on GLUT1 trafficking observed with ATG5 KO vs. KOs in other membrane atg8ylation genes, which were significant in all experiments but presented effects lower in magnitude than ATG5 KO (*Figures 4 and 6*). One possibility was that since ATG3, ATG5, and ATG7 all participate in the same process of membrane atg8ylation, that our KO mutants had unequal gene inactivation levels. This was ruled out by comparing effects of ATG3 KO, ATG5 KO, and ATG7 KO cells for their effects on LC3 puncta formation in a well-defined system of starvation-induced autophagy. The results of these analyses indicated that all three KOs had indistinguishable effects on LC3 puncta formation during canonical autophagy induction by starvation (*Figure 8—figure supplement 3A, B*). Thus, the stronger effects of ATG5 KO on retromer vis-à-vis KOs in other atg8ylation genes reflected additional action of ATG5. In the absence of ATG5, less VPS26 and VPS35 could be pulled down in co-IPs with YFP-VPS29 (*Figure 8—figure supplement 2C–F*). We conclude that ATG5, in addition to contributing to the retromer-dependent sorting via membrane atg8ylation and its downstream effector mechanisms which maintain healthy endolysosomal organelles, has additional effects on the retromer.

## Discussion

In this study, we have uncovered a new role of ATG5 and membrane atg8ylation machinery beyond the conventional function in canonical autophagy and the hitherto recognized noncanonical processes. ATG5 interacts with the retromer complex, whereas ATG5 and membrane atg8ylation affect retromer function in sorting of its cognate cargo GLUT1. Inactivation of ATG5 and of other membrane atg8ylation genes but not of the genes involved in canonical autophagy perturbs retromer-dependent trafficking. ATG5 and membrane atg8ylation display dual and combined effects on retromer function. The first one is through endolysosomal membrane maintenance ensuring proper cargo sorting by the retromer complex and its adaptor SNX27. The second one is via an association of ATG5 with the retromer complex. These and other activities of ATG5 underly the unique and often perplexing

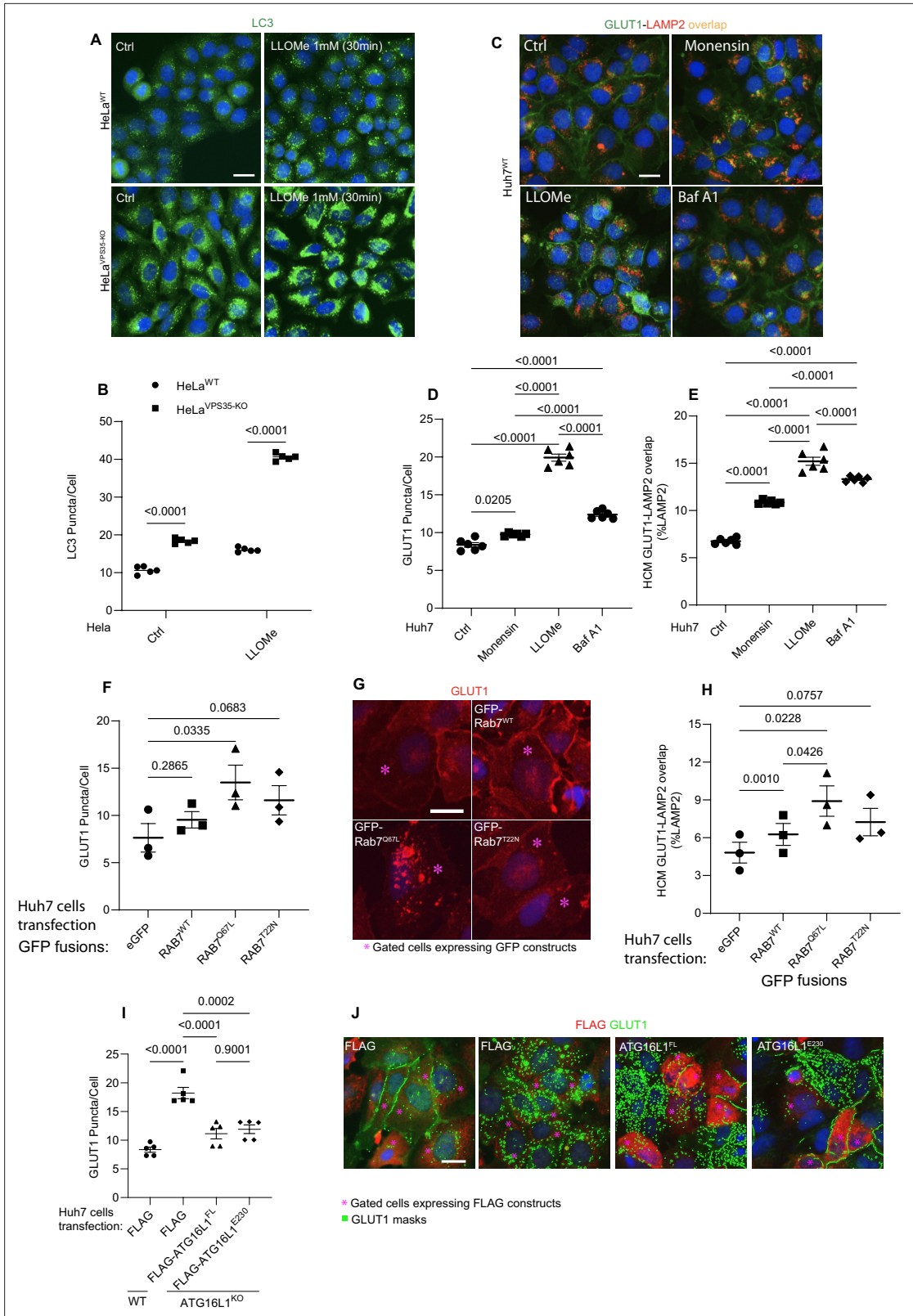

**Figure 7.** Lysosomal perturbations cause GLUT1 mis-sorting. (**A, B**) High content microscopy (HCM) imaging and quantification of LC3 response (puncta/cell of endogenous LC3 immunofluorescent profiles) in HeLa[WT], and HeLa[VPS35-KO] cells in response to lysosomal damage by LLOMe (1 mM, 30 and 60 min). Scale bar, 20 μm. Data, means ± SE (*n* = 5), one-way ANOVA with Tukey's multiple comparisons. (**C–E**) HCM quantification of GLUT1 response (puncta/cell of endogenous GLUT1) (**D**) and its localization to endolysosomal compartments (% of LAMP1 profiles positive for GLUT1

*Figure 7 continued*

immunostaining) (**E**) in Huh7$^{WT}$ treated with or without Monensin (100 μM), LLOMe (100 μM), and Bafilomycin A1 (100 nM) for 45 min. Scale bar, 20 μm. Data, means ± SE ($n$ = 5), one-way ANOVA with Tukey's multiple comparisons. Analysis of GLUT1 puncta/cell (**F, G**) and its localization to endolysosomal compartments (% of LAMP1 profiles positive for GLUT1 immunostaining), (**H**) phenotype monitored by HCM quantification in Huh7$^{WT}$ cells transfected with GFP (control) or GFP-Rab7$^{WT}$, GFP-Rab7$^{Q67L}$, and GFP-Rab7$^{T22N}$, expressing plasmids. Scale bar, 20 μm. Data, means ± SE ($n$ = 3); one-way ANOVA with Tukey's multiple comparisons. Analysis of GLUT1 puncta/cell (**I, J**) by HCM quantification in Huh7 WT and ATG16L1-KO cells complemented with Flag (control), Flag-ATG16L1$^{FL}$, or Flag-ATG16L1$^{E230}$, expressing plasmids. Scale bar, 20 μm. Data, means ± SE ($n$ = 5); one-way ANOVA with Tukey's multiple comparisons. HCM images in all relevant panels, examples from a bank of unbiased operator-independent machine-collected and algorithm-processed fields containing a minimum of 500 primary objects/cells per well (5 wells minimum per 96-well plate; 3 plates minimum), per cell line/condition.

The online version of this article includes the following source data and figure supplement(s) for figure 7:

**Source data 1.** Numerical values for quantification in graphs.

**Figure supplement 1.** CASM agonists effects on retromer cargo GLUT1 in the absence of changes in TBC1D5-LC3A association.

**Figure supplement 1—source data 1.** PDF files containing original immunoblots for *Figure 7—figure supplement 1* indicating relevant bands.

**Figure supplement 1—source data 2.** Original files for immunoblots displayed in *Figure 7—figure supplement 1*.

**Figure supplement 1—source data 3.** Numerical values for quantification in graphs.

phenotypic manifestations of ATG5 observed in cells and in murine infection models (*Wang et al., 2023*; *Castillo et al., 2012*; *Virgin and Levine, 2009*; *Watson et al., 2012*; *Kimmey et al., 2015*; *Golovkine et al., 2023*; *Kinsella et al., 2023*), which served as the initial impetus for the present study. More generally, the finding that membrane atg8ylation influences retromer function further expands the range of this process specializing in homeostatic responses to membrane stress, damage, and remodeling signals (*Deretic and Lazarou, 2022*; *Deretic et al., 2024*; *Kumar et al., 2021b*).

Mirroring the role of membrane atg8ylation on retromer function, a loss of VPS35 brings about LyHYP and influences cellular responses to lysosomal damage similarly to the inactivation of ATG5. The parallel between retromer and ATG5 is not perfect, as VPS35 knockout does not divert membrane repair protein ALIX from damaged lysosomes. This can be explained by sequestration of ALIX by the alternative conjugation complex ATG12-ATG3 formed in the absence of ATG5 (*Wang et al., 2023*). The latter phenomenon renders ALIX unavailable for lysosomal repair (*Wang et al., 2023*). VPS35 knockout has no effects on ALIX, and hence the differences between ATG5 and VPS35 in the ALIX component of LyHYP. Nevertheless, inactivation of either ATG5 or VPS35 confers similarly heightened Gal3 (*Jia et al., 2020b*; *Aits et al., 2015*) and ubiquitin (*Papadopoulos et al., 2017*) responses as hallmarks of elevated lysosomal damage.

A further connectivity between the retromer system and membrane atg8ylation was observed at the level of LC3 puncta formation elicited by noncanonical triggers such as LLOMe treatment, which induces CASM and lysosomal repair. The effects of VPS35 on lysosomal functionality and morphology have been previously noted (*Cui et al., 2019*; *Daly et al., 2023*), and have been ascribed to the effects of the retromer complex on the recycling of the sorting receptors for lysosomal hydrolases, such as cation-independent mannose 6-phosphate receptor (CI-MPR) (*Cui et al., 2019*). We have detected significant increase in LC3 puncta in VPS35 KO cells stimulated for autophagy by starvation, consistent with the reported diminished lysosomal degradative capacity when VPS35 is absent, an effect ascribed to aberrant CI-MPR sorting (*Cui et al., 2019*; *Daly et al., 2023*). Whether CI-MPR is sorted by retromer or independently by the ESCPE-I complex (*Simonetti et al., 2019*) remains controversial (*Seaman, 2021*; *Simonetti et al., 2019*; *Buser and Spang, 2023*; *Cui et al., 2019*; *Kvainickas et al., 2019*; *Kvainickas et al., 2017*; *Seaman, 2007*; *Simonetti et al., 2017*). This controversy has been extended to the sorting of cation-dependent mannose 6-phosphate receptor (CD-MPR) (*Cui et al., 2019*; *Buser et al., 2022*), further confounded by the fact that retriever, which participates in receptor recycling, shares VPS29 subunit with the retromer (*McNally et al., 2017*). Our data showing perturbances in CI-MPR sorting in ATG5 KO and VPS35 KO cells can be best explained as indirect effects via ATG5 and VPS35 roles in endolysosomal homeostasis, which in turn may affect multiple sorting complexes.

In this study, we observed a miss-localization of the small GTPase Rab7 to lysosomes in ATG5 KO cells, paralleling the effects on GLUT1. Retromer and SNX adaptors (SNX3) interact with Rab7 (*Harrison et al., 2014*). Rab7 affects retromer-dependent sorting (*Roy et al., 2017*; *Harrison et al.,*

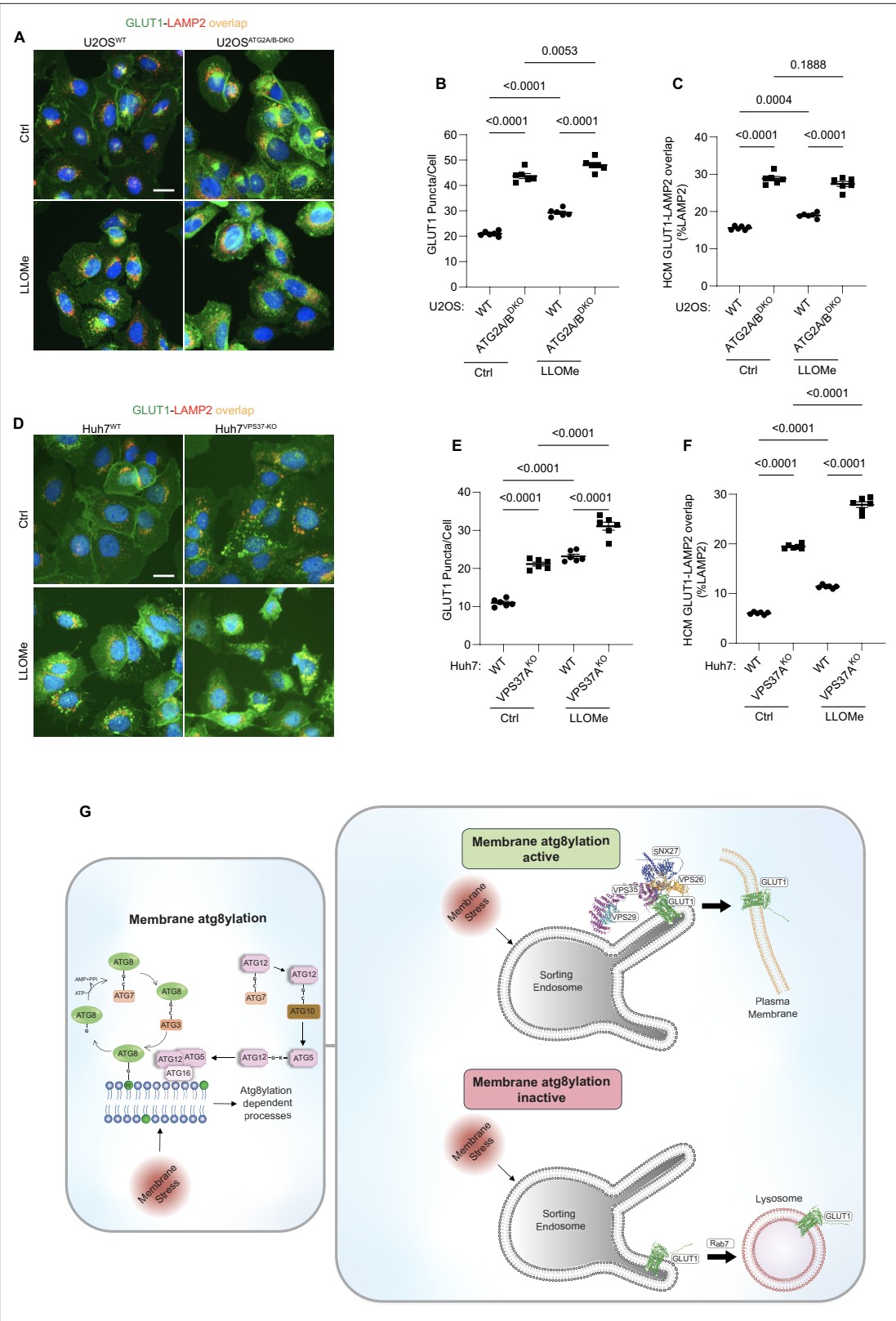

**Figure 8.** Membrane atg8ylation and endolysosomal homeostasis affect retromer-dependent sorting. (**A–C**) High content microscopy (HCM) quantification of GLUT1 (immunostaining of endogenous protein) puncta/cell (**B**) and GLUT1 colocalization with LAMP2 (% of LAMP2 profiles positive for GLUT1; overlap area), (**C**) in U2OS[WT] and U2OS[ATG2A/B-DKO] cells treated with or without LLOMe (100 µM) for 45 min. Scale bar, 20 µm. Data, means ± SE (*n* = 6); one-way ANOVA with Tukey's multiple comparisons. (**D–F**) HCM quantification of GLUT1 (immunostaining of endogenous protein) puncta/cell

*Figure 8 continued*

(**E**) and GLUT1 colocalization with LAMP2 (% of LAMP2 profiles positive for GLUT1; overlap area), (**F**) in Huh7[WT] cand Huh7[VPS37A-KO] cells treated with or without LLOMe (100 μM) for 45 min. Scale bar, 20 μm. Data, means ± SE (*n* = 6); one-way ANOVA with Tukey's multiple comparisons. HCM images in all relevant panels, examples from a bank of unbiased operator-independent machine-collected and algorithm-processed fields containing a minimum of 500 primary objects/cells per well (5 wells minimum per 96-well plate; 3 plates minimum), per cell line/condition. (**G**) Schematic: Membrane atg8ylation maintains membrane homeostasis under basal or stress conditions and, independently of canonical autophagy, affects retromer function. A functional atg8ylation apparatus is required for proper sorting of the retromer cargo GLUT1.

The online version of this article includes the following source data and figure supplement(s) for figure 8:

**Source data 1.** Numerical values for quantification in graphs.

**Figure supplement 1.** High content microscopy (HCM) images of GLUT1.

**Figure supplement 2.** Effects of ATG5 knockout on MPR sorting.

**Figure supplement 2—source data 1.** Numerical values for quantification in graphs (extrapolated from the graph by plot digitizer).

**Figure supplement 3.** ATG5 affects retromer assembly.

**Figure supplement 3—source data 1.** PDF files containing original immunoblots for *Figure 8—figure supplement 3* indicating relevant bands.

**Figure supplement 3—source data 2.** Original files for immunoblots displayed in *Figure 8—figure supplement 3*.

**Figure supplement 3—source data 3.** Numerical values for quantification in graphs.

*2014*; *Jimenez-Orgaz et al., 2018*; *Seaman et al., 2018*). Depletion of Rab7 reduces endosomal association of retromer (*Harrison et al., 2014*) whereas retromer status affects Rab7, that is, in cells with retromer subunits knocked out, Rab7 anomalously accumulates on lysosomes (*Jimenez-Orgaz et al., 2018*). Consistent with this, we observed that ATG5 KO phenocopied effects of VPS35 KO and caused Rab7 re-localization to lysosomal membranes. Rab7, like other Rabs (*Borchers et al., 2021*), is controlled by GEFs (Mon1-Cz1), GAPs (ArmusTBC1D2A, TBC1D2B, TBC1D15), and GDI (*Stroupe, 2018*). It is curious that all Rab7 GAPs have LC3 interaction region (LIR) motifs and bind mATG8s: Armus/TBC12DA and TBC1D5 bind LC3A (*Roy et al., 2017*; *Popovic et al., 2012*), GABARAPL1 (*Popovic et al., 2012*) and LC3C (*Roy et al., 2017*); TBCD12B binds all mATG8s *Popovic et al., 2012*; *Behrends et al., 2010*; and TBC1D15 binds LC3A (*Yamano et al., 2014*), LC3B, LC3C, GABARAP, GABARAP L1, and GABARAPL2 (*Behrends et al., 2010*; *Yamano et al., 2014*). Prior elegant studies have shown that genetic ablation of the Rab7 GAP TBCD15 results in hyperactivation of Rab7 in a wrong intracellular locale (lysosome) (*Jimenez-Orgaz et al., 2018*). In our work, carried out under basal conditions or in cells subjected to lysosomal damage, we observed both translocation of Rab7 to and entrapment of GLUT1 on lysosomes. However, we did not detect any changes in TBC1D5–mATG8s interactions, suggesting that sequestration of this Rab7 GAP by mATG8s, leading to increased plasma membrane localization of GLUT1 under glucose starvation conditions (*Roy et al., 2017*), is not at play under basal and endolysosmal damage or luminal alkalinization conditions tested here. We did not test glucose limitation or conditions that induce canonical autophagy or engage more complex metabolic pathways. Furthermore, we found that overexpression of constitutively active Rab7 in cells grown in glucose/nutrient-rich media causes lysosomal retention of GLUT1. This is consistent with the reports by others that perturbances in the retromer-TBC1D5 complex lead to miss-localization of Rab7 to more lysosomal-like subdomains within the endosomal system (*Kvainickas et al., 2019*) and that inactivation of Rab7's GAP TBC1D5 leads to entrapment within the endosomal system of the plasma membrane receptors as well as receptors that normally cycle back to the trans-Golgi network (*Jia et al., 2016*).

The role of ATG5 in retromer-dependent sorting of GLUT1 appears to have two components, the first one reflects ATG5 being a part of the E3 ligases for membrane atg8ylation and the second one reflects ATG5's being in protein complexes with the core retromer subunits VPS26, VPS29, and VPS35. The latter association potentially explains the stronger effects of ATG5's absence on GLUT1 sorting relative to the inactivation of other membrane atg8ylation genes. Of note, interactions between ATG5 and retromer are independent of the conjugation status of ATG5, as both ATG12—ATG5 conjugates and unconjugated ATG5 were found in protein immunoprecipitates of retromer components. This is consistent with our LC–MS/MS proteomic data indicating a conjugation status-independent increase in the proximity of ATG5 and retromer subunits during lysosomal damage. One consequence of the association between retromer and ATG5, a key component of the membrane atg8ylation apparatus, is that this could spatially direct proper membrane atg8ylation and be in part responsible for the effects

of VPS35 KO on lysosomal quality and sensitivity to membrane stress or injury. In our hands, Alpha-Fold modeling of ATG5 with retromer components did not yield any high-confidence structures, and the best of the low-to-moderate probability structures when subjected to mutational analysis did not result in disruption of ATG5–retromer association observed by co-IPs (data not shown). Identification of the ATG5 partners that bring it to the vicinity of the retromer is yet to be accomplished and remains one of the limitations of the present study.

We used GLUT1 as a well-defined cargo sorted by the retromer in conjunction with SNX27 (*Gallon et al., 2014*; *Steinberg et al., 2013*). SNX27 is a versatile adapter, linking retromer to and controlling endosome-to-plasma membrane recycling of nearly 80 proteins such as signaling receptors, ion channels, amino acid and other nutrient transporters (*Steinberg et al., 2013*). SNX27 was also missorted in ATG5 deficient cells. This suggests that knockouts in ATG5 and in other membrane atg8ylation genes could have complex pleiotropic effects on cellular functions. Given that ATG5 has a particularly strong impact on retromer-dependent sorting, this could in part explain its unique role in the exquisite susceptibility of Atg5 mutant mice to experimental *M. tuberculosis* infections. Such effects likely contribute to the inflammatory action of neutrophils observed in Atg5^fl/fl^ LysM-Cre mice (*Wang et al., 2023*; *Castillo et al., 2012*; *Watson et al., 2012*; *Kimmey et al., 2015*; *Golovkine et al., 2023*; *Kinsella et al., 2023*). Specifically, ATG5 KO-dependent GLUT1 miss-sorting may be one of the contributors to increased *M. tuberculosis* pathogenesis in infection sites. In the context of tuberculosis, diabetes, which includes glucose uptake dysregulation, is associated with increased incidence of active disease and adverse outcomes (*Dheda and Maartens, 2016*; *Dooley and Chaisson, 2009*). More generally, glucose uptake and metabolism are highly important for normal physiology and in various disease states including immune responses to infection, cancer, neurodegeneration, cardiovascular health, and metabolic disorders. We postulate that deficiencies or polymorphisms in membrane atg8ylation genes could have contributory roles in health and disease via GLUT1 and additional transporter proteins and signaling receptors that are dependent on retromer for their proper positioning and function in the cell.

One of the limitations of our study is that beyond the effect of membrane atg8ylation on quality of lysosomal membrane and its homeostasis, there could be more direct effects on retromer that still need to be understood. Another limitation of our study is that we have focused on basal conditions or conditions causing lysosomal damage, whereas metabolic stress including glucose excess or limitation with its multitude of metabolic effects have not been addressed. Nevertheless, we find that the membrane atg8ylation and retromer systems are intertwined, and that they affect each other's biological outputs, one being resilience of lysosomes to basal stress or induced damage (*Figure 8G*) and the other being endosomal-plasma membrane protein sorting, both being of fundamental interest and potential therapeutic value.

# Materials and methods
## Cells and cell line models
HEK293T and HeLa cells were from ATCC (American Type Culture Collection). Huh7 cells were from Rocky Mountain Laboratories. HeLa HEXA and TKO cell lines and their parental cell line were from Michael Lazarou. U2OS and its ATG2A/B DKO derivative were from Fulvio Reggiori.

## Mice
Atg5^fl/fl^ LysM-Cre^-^, Atg5^fl/fl^ LysM-Cre^+^ mice were previously described (*Castillo et al., 2012*; *Manzanillo et al., 2012*).

## Housing and husbandry conditions of experimental animals
All mice were housed in AAALAC-accredited Animal Research Facility (ARF) of the University of New Mexico Health Sciences Center (UNM-HSC) and institutionally approved husbandry conditions and approved breeding protocols were followed 23-201351-B-HSC. *M. tuberculosis*-infected animals were housed in a separate ABSL3 suite within the UNM HSC ARF facility and all staff followed strict ABSL3, BSL3, and animal protocols approved by the UNM HSC Biosafety Committee and the Institutional Animal Care and Use Committee. The protocol number was 23-201379-HSC. The study was compliant with all relevant ethical guidelines for animal research.

## Antibodies

Antibodies from Abcam were ATG5 (1:2000 for western blot (WB), ab108327), ATG7 (1:2000 for WB, ab52472), GFP (1:2000 for WB; 1:300 for immunoprecipitation (IP), ab290), mCherry (1:1000 for WB, ab183628), VPS35 (1:500 for WB; 1:1000 for IF, ab10099), VPS29 (1:500 for immunofluorescence (IF), ab10160), GLUT1 (1:3000 western blot (WB), 1:1000 for immunofluorescence (IF), ab115730), CI-M-PR(2G11) (2 µg/ml for antibody uptake assay, ab2733), RAB7 (1:2000 western blot (WB), and 1:500 for immunofluorescence (IF), ab137029).

Antibodies from Biolegend were ALIX (1:400 for immunofluorescence (IF), #634502) and Galectin-3 (1:200 for Immunofluorescence (IF), #125402).

Antibodies from Proteintech were VPS26A (1:500 for western blot (WB), 12804-1-AP), VPS35 (1:500 for western blot (WB), 1:1000 for immunofluorescence (IF), 10236-1-AP), and TBC1D5 (1:1000 for western blot (WB), 17078-1-AP).

Antibodies from Sigma-Aldrich were ATG3 (1:1000 for western blot (WB), #A3231), Ubiquitin (FK2, 1:500 for immunofluorescence (IF), 04-263), and mouse Anti-FLAG M2 (1:500 for immunofluorescence (IF), #F1804).

Antibodies from Cell Signaling were α-Tubulin (DM1A) (1:3000 for western blot (WB); #3873) and LAMP1 (1:3000 for immunofluorescence (IF), #9091).

Other antibodies used in this study were from the following sources: beta-Actin (1:500 for western blot (WB), sc-47778) and GAPDH (1:500 for western blot (WB), sc-47724) from Santa Cruz Biotechnology; LAMP2 (1:1000 for immunofluorescence (IF), H4B4) from DSHB of University of Iowa; ATG5 (1: 500 for western blot (WB), ASA-B0113) from Novateinbio, SNX27 (1:1000 for western blot (WB), MA5-27854) from Invitrogen.

Secondary antibodies labeled with Alexa Fluor 488, 568, 647 (1:500 for immunofluorescence (IF)) and IgG-HRP (1:10,000 for western blots (WB)) were from Thermo Fisher Scientific. IgG Polyclonal Antibody Goat anti-mouse IRDye 680 (LI-COR, 925-68020), and Goat anti-rabbit IRDye 800 (LI-COR, 926-32211) secondary antibodies were from LI-COR Biosciences.

## Reagents and antibiotics

Bafilomycin A1 (BafA1, InvivoGen; 13D02-MM), Monensin (Sigma, M5273), LLOMe (Sigma, L7393), Lipofectamine 2000 (Thermo Scientific, 11668019); Triton X-100 (OmniPur, 9410-OP), saponin (Sigma, S4521-25G). DMEM (Gibco, #11995040), and Penicillin–Streptomycin (1000 U/ml; Gibco, #15140122). OptiMEM from Life Technologies, Puromycin dihydrochloride (Sigma, P9620), Hygromycin B (Sigma, H0654).

## Plasmids and transfection

Plasmids used in this study, such as ATG5 were generated by first cloning inserts into pDONR221 (Gateway Technology cloning vector, Thermo Scientfic) using a BP cloning reaction and the expression vectors were made utilizing LR cloning reaction (Gateway, Thermo Fisher) in appropriate (pDEST) destination vectors for immunoprecipitation assay. Addgene clones were: eGFP-Rab7 WT (Addgene, #12605), eGFP-Rab7$^{Q67L}$ (Addgene, #28049), and eGFP-Rab7$^{T22N}$ (Addgene, #28049). Additional plasmids were pDEST-3X flag (from Terje Jonansen), Flag-Atg16l1$^{FL}$, and Flag-Atg16l1$^{E230}$ (synthetic clone prepared by Thabata Duque). Plasmid transfections were performed using the Lipofectamine 2000/3000 Transfection Reagent (Thermo Fisher Scientific, #11668019).

## siRNAs

The siRNAs were from Horizon Discovery (formerly known as Dharmacon): siGENOME Non-Targeting Control siRNA (Identifier: D-001810-01-05, Target sequence: UGGUUUACAUGUC-GACUAA); siGENOME human ATG5 SMARTpool siRNA (Identifier: M-004374-04-0005). ATG5 was a pool of four different siRNAs targeting a single gene with individual siRNA sequences: GGAAU-AUCCUGCAGAAGAA; CAUCUGAGCUACCCGGAUA; GACAAGAAGACAUUAGUGA; and CAAUUGGUUUGCUAUUUGA.

## Cornell model of *M. tuberculosis* latent infection

For this study, a total of 68 mice were used (35 Atg5^fl/fl LysM-Cre^+ and 33 Atg5^fl/fl LysM-Cre^-). Early in the course of the study, one mouse died due to malocclusion and thus could not be included in the final analysis.

Inoculum was prepared by diluting *M. tuberculosis* Erdman frozen stock 1:50 in PBS/0.01% Tween for a final amount of ~7.38e$^6$ CFU/ml. Inoculum was serially diluted five times at 1:10 each time in PBS/ Tween. 50 µL aliquots of the third, fourth, and fifth dilutions were plated on 7H11 agar plates to determine actual inoculum CFUs (Actual inoculum for this study: 6.45e$^6$ CFU/ml). Remaining inoculum was added to Glas-Col inhalation System and mice were infected via aerosol according to the following machine settings:

Glas-Col Cycle Settings:
*M. tuberculosis* Erdman diluted 1:50, targeting 200 CFU/mouse.

1. Preheat (15 min)
2. Nebulizing (20 min)
3. Cloud decay (20 min)
4. Decontamination (15 min) – UV lights ON
5. Cool down period (10 min)

Immediately following aerosol infection, three C57BL/6 mice, infected in parallel with the experimental cohort, were euthanized to determine lung deposition CFUs. 5 × 200 µl aliquots of neat, homogenized tissue were grown for 2–3 weeks at 37°C and 5% $CO_2$. Initial deposition: ca. 100 CFUs per lung. This procedure was used for all subsequent CFU determinations in this study.

After a period of 2.5 weeks bacterial growth was assessed by determining lung CFUs (2 Cre^+ and 1 Cre^- sacrificed).

Subsequently, mice were treated PO with antibiotics (0.1 g/l INH and 0.15 g/l RIF) in drinking water for 8 weeks and bacterial clearance by chemotherapy was assessed by determining lung CFUs (11 Cre^+ and 10 Cre^- sacrificed).

The remaining mice were subjected to antibiotic washout for 7 weeks plus a spontaneous reactivation period with no treatment for 3 weeks at which time lung CFUs were determined (8 Cre^+ and 8 Cre^- sacrificed).

After washout and reactivation periods, dexamethasone was administered by IP injection 5 times/ week at 0.08 mg/mouse/day to induce immunosuppression. Mice were immunosuppressed in this manner for 6 weeks after which point lung CFUs were determined (14 Cre^+ and 13 Cre^- sacrificed).

## Generation of CRISPR mutant cells

Knockout cells (HeLa^ATG5-KO, HeLa^VPS35-KO Huh7^ATG5-KO, Huh7^ATG3-KO, Huh7^ATG7-KO, Huh7^VPS35-KO) were generated by CRISPR/Cas9-mediated knockout system. The lentiviral vector lentiCRISPRv2 carrying both Cas9 enzyme and a gRNA targeting ATG5 (gRNA-puro: AAGAGTAAGTTATTTGACGT), ATG3 (gRNA-Hygro: GTGAAGGCATACCTACCAAC), ATG7 (gRNA-puro: CTTCCGTGACCGTACCATGC), VPS35 (gRNA-hygro: GCTCACCGTGAAGATGGACC), or Scramble (gRNA-hygro: GTGTAGTTCGACCATT CGTG) were transfected into HEK293T cells together with the packaging plasmids psPAX2 and pCMV-VSV-G at the ratio of 5:3:2. Two days after transfection, the supernatant containing lentiviruses was collected. Cells were infected by the lentiviruses with 8–10 µg/ml polybrene. 36 hr after infection, the cells were selected with puromycin (1–10 µg/ml) or hygromycin (100–500 µg/ml) for 1 week in order to select knockout cells. All knockouts were confirmed by Western blot. Selection of single clones was performed by dilution in 96-well.

## High content microscopy

Cells in 96-well plates were fixed in 4% paraformaldehyde for 5 min. Cells were then permeabilized with 0.1% saponin in 3% bovine serum albumin (BSA) for 30 min followed by incubation with primary antibodies for 2 hr and secondary antibodies for 1 hr. Hoechst 33342 staining was performed for 3 min. High content microscopy with automated image acquisition and quantification was carried out using a Cellomics HCS scanner and iDEV software (Thermo Fisher Scientific). Automated epifluorescence image collection was performed for a minimum of 500 cells per well. Epifluorescence images were machine analyzed using preset scanning parameters and object mask definitions. Hoechst 33342

staining was used for autofocusing and to automatically define cellular outlines based on background staining of the cytoplasm. Primary objects were cells, and regions of interest (ROIs) or targets were algorithm-defined by shape/segmentation, maximum/minimum average intensity, total area and total intensity, etc., to automatically identify puncta or other profiles within valid primary objects. Each experiment (independent biological repeats; $n \geq 3$) consists of machine-identified 500 valid primary objects/cells per well, $\geq 5$ wells/sample. All data collection, processing (object, ROI, and target mask assignments) and analyses were computer driven independently of human operators. HCM also provides a continuous variable statistic since it does nor rely on parametric reporting cells as positive or negative for a certain marker above or below a puncta number threshold.

### Antibody uptake assay

CI-MPR antibody was either incubated with cell for 1 hr in 37°C incubator or pulse-chase with the following procedures: cells were preincubate at 4°C for 15 min, followed by 30 min incubation with antibody at 4°C, three times wash with PBS, and 30 min incubation in 37°C incubator. The cells were then fixed and subject to following procedure.

### Co-IP and immunoblotting assays

For co-IP, cells transfected with 8–10 µg of plasmids were lysed in ice-cold NP-40 buffer (Thermo Fisher Scientific) supplemented with protease inhibitor cocktail (11697498001; Roche) and 1 mM PMSF (93482; Sigma-Aldrich) for 30 min on ice. Lysates were centrifuged for 10 min at 10,000 × $g$ at 4°C. Supernatants were incubated with (2–3 µg) antibodies overnight at 4°C. The immune complexes were captured with Dynabeads (Thermo Fisher Scientific), followed by three times washing with 1× PBS. Proteins bound to Dynabeads were eluted with 2× Laemmli sample buffer (Bio-Rad) and subjected to immunoblot analysis.

For immunoblotting, lysates were centrifuged for 10 min at 10,000 × $g$ at 4°C. Supernatants were then separated on 4–20% Mini-PROTEAN TGX Precast Protein Gels (Bio-Rad) and transferred to nitrocellulose membranes. Membranes were blocked in 3% BSA for 1 hr at RT and incubated over-night at 4°C with primary antibodies diluted in blocking buffer. They were then incubated with an HRP-conjugated secondary antibody, and proteins were detected using ECL and developed using ChemiDoc Imaging System (Bio-Rad). Analysis and quantification of bands were performed using ImageJ software.

### Lysosomal purification by LysoIP

Plasmids for LysoIP were purchased from Addgene. Cells were plated in 10 cm dishes in DMEM and 10% fetal bovine serum and transfected with pLJC5-TMEM192-3xHA or pLJC5-TMEM192-2XFLAG constructs at 75–85% confluency. After 24 hr of transfection, cells with or without treatment were quickly rinsed twice with PBS and then scraped in 1 ml of KPBS (136 mM KCl, 10 mM KH$_2$PO$_4$, pH 7.25 adjusted with KOH) and centrifuged at 3000 rpm for 2 min at 4°C. Pelleted cells were resuspended in 1000 µl KPBS and reserved 50 µl (for the whole cell lysate) before further processing. The remaining cells were gently homogenized with 20 strokes of a 2-ml homogenizer. The homogenate was then centrifuged at 3000 rpm for 2 min at 4°C and the supernatant was incubated with 100 µl of KPBS prewashed anti-HA magnetic beads (Thermo Fisher) on a gentle rotator shaker for 15 min. Immu-noprecipitants were then gently washed three times with KPBS and eluted with 2× Laemmli sample buffer (Bio-Rad) and subjected to immunoblot analysis.

### Immunofluorescence confocal microscopy and analysis

Cells were plated onto coverslips in 6-well plates. After treatment, cells were fixed in 4% paraformal-dehyde for 5 min followed by permeabilization with 0.1% saponin in 3% BSA for 30 min. Cells were then incubated with primary antibodies for 2 hr and appropriate secondary antibodies Alexa Fluor 488 or 568 (Thermo Fisher Scientific) for 1 hr at room temperature. Coverslips were mounted using Prolong Gold Antifade Mountant (Thermo Fisher Scientific). Images were acquired using a confocal microscope (META; Carl Zeiss) equipped with a 63 3/1.4 NA oil objective, camera (LSM META; Carl Zeiss), and AIM software (Carl Zeiss).

### Sample preparation for LC–MS/MS

The previously described HeLaFlp-In-APEX2-ATG5-WT and HeLaFlp-InAPEX2-ATG5-K130R cells (*Wang et al., 2023*) were incubated in complete medium supplemented with 500 µM biotin–phenol

(AdipoGen) with or without 2 mM LLOMe for 30 min. A 1-min pulse with 1 mM $H_2O_2$ at room temperature was stopped with quenching buffer (10 mM sodium ascorbate, 10 mM sodium azide and 5 mM Trolox in PBS). All samples were washed twice with quenching buffer, and twice with PBS for 1 min. For LC–MS/MS analysis, cell pellets were lysed in 500 µl ice-cold lysis buffer (6 M urea, 0.3 M NaCl, 1 mM EDTA, 1 mM EGTA, 10 mM sodium ascorbate, 10 mM sodium azide, 5 mM Trolox, 1% glycerol and 25 mm Tris–HCl, pH 7.5) for 30 min by gentle pipetting. Lysates were clarified by centrifugation and protein concentrations were determined using Pierce 660 nm protein assay reagent. Streptavidin-coated magnetic beads (Pierce) were washed with lysis buffer. A total of 1 mg of each sample was mixed with 100 µl of streptavidin beads. The suspensions were gently rotated at 4°C overnight to bind biotinylated proteins. The flow-through after enrichment was removed and the beads were washed in sequence with 1 ml IP buffer (150 mM NaCl, 10 mM Tris–HCl, pH 8.0, 1 mM EDTA, 1 mM EGTA, 1% Triton X-100) twice; 1 ml 1 M KCl; 1 ml of 50 mM $Na_2CO_3$; 1 ml 2 M urea in 20 mM Tris–HCl (pH 8.0); and 1 ml IP buffer. Biotinylated proteins were eluted and processed for mass spectrometry. Protein samples on magnetic beads were washed four times with 200 µl of 50 mM Triethyl ammonium bicarbonate (TEAB) with a 20-min shake time at 4°C in between each wash. Roughly 2.5 µg of trypsin was added to the bead and TEAB mixture and the samples were digested over night at 800 rpm shake speed. After overnight digestion the supernatant was removed, and the beads were washed once with enough 50 mM ammonium bicarbonate to cover. After 20 min at a gentle shake the wash is removed and combined with the initial supernatant. The peptide extracts are reduced in volume by vacuum centrifugation and a small portion of the extract is used for fluorometric peptide quantification (Thermo scientific Pierce). One microgram of sample based on the fluorometric peptide assay was loaded for each LC–MS analysis.

## Liquid chromatography–tandem mass spectrometry

Peptides were desalted and trapped on a Thermo PepMap trap and separated on an Easy-spray 100 µm × 25 cm C18 column using a Dionex Ultimate 3000 nUPLC at 200 nl/min. Solvent A = 0.1% formic acid, Solvent B = 100% acetonitrile 0.1% formic acid. Gradient conditions = 2% B to 50% B over 60 min, followed by a 50–99% B in 6 min and then held for 3 min than 99% B to 2% B in 2 min and total run time of 90 min using Thermo Scientific Fusion Lumos mass spectrometer. The samples were run in DIA mode; mass spectra were acquired using a collision energy of 35, resolution of 30 K, maximum inject time of 54ms and an AGC target of 50 K, using staggered isolation windows of 12 Da in the *m/z* range 400–1000 *m/z*.

## DIA quantification and analysis

DIA data were analyzed using Spectronaut 14.10 (Biognosys Schlieren, Switzerland) using the directDIA workflow with the default settings. Briefly, protein sequences were downloaded from Uniprot (Human Proteome UP000005640), ATG5 from Uniprot and common laboratory contaminant sequences from https://thegpm.org/crap/. Trypsin/P specific was set for the enzyme allowing two missed cleavages. Fixed Modifications were set for Carbamidomethyl, and variable modification were set to Acetyl (Protein N-term) and Oxidation. For DIA search identification, PSM and Protein Group FDR was set at 1%. A minimum of 2 peptides per protein group were required for quantification. Proteins known to be endogenously biotinylated were excluded from consideration.

## Quantification and statistical analysis

Data in this study are presented as means ± SEM ($n \geq 3$). Data were analyzed with either ANOVA with Tukey's HSD post hoc test, or a two-tailed Student's *t*-test. For HCM, $n \geq 3$ (independent experiments carried out on different 96-well plates). For each well in 96-well plates, ≥500 valid primary objects/cells were imaged and analyzed, with ≥5 wells per plate per sample. Statistical significance: $p \geq 0.05$ (not significant); <0.05 (significant). For HCM, sample size was based on a historic power analysis (published studies), with large effect size (differences and variability derived from published work), power 80%, β 20%, and α 5%, assuming normal distribution and favoring type II false-negative errors over type I false-positive errors. Band intensity in immunoblots, $n = 3$ (biological replicates); no power analysis was performed.

## Acknowledgements

We thank Ryan Peters and Seong Won Choi for carrying out animal infection studies, and Fulvio Reggiori for U2OS ATG2 knockout derivatives. This work was supported by NIH grants R37AI042999 and R01AI111935, and center grant P20GM121176 to VD. Mass spectrometry analysis, BP, and MS were supported by NIH shared instrumentation grant S10OD021801.

## Additional information

### Funding

| Funder | Grant reference number | Author |
|--------|----------------------|--------|
| National Institute of Allergy and Infectious Diseases | R37AI042999 | Vojo Deretic |
| National Institute of Allergy and Infectious Diseases | R01AI111935 | Vojo Deretic |
| National Institute of General Medical Sciences | P20GM121176 | Vojo Deretic |
| NIH shared instrumentation | S10OD021801 | Brett S Phinney |

The funders had no role in study design, data collection and interpretation, or the decision to submit the work for publication.

### Author contributions

Masroor Ahmad Paddar, Conceptualization, Formal analysis, Validation, Investigation, Visualization, Methodology, Writing – original draft, Writing – review and editing; Fulong Wang, Conceptualization, Formal analysis, Investigation, Methodology, Writing – original draft; Einar S Trosdal, Conceptualization, Investigation, Visualization, Methodology; Emily Hendrix, Formal analysis; Yi He, Michal Mudd, Investigation; Michelle R Salemi, Investigation, Methodology; Jingyue Jia, Resources; Thabata Duque, Ruheena Javed, Conceptualization, Resources; Brett S Phinney, Formal analysis, Investigation; Vojo Deretic, Conceptualization, Resources, Formal analysis, Supervision, Funding acquisition, Investigation, Methodology, Writing – original draft, Project administration, Writing – review and editing

### Author ORCIDs

Masroor Ahmad Paddar ⓘ https://orcid.org/0009-0008-5639-3006
Vojo Deretic ⓘ https://orcid.org/0000-0002-3624-5208

### Ethics

All mice were housed in AAALAC-accredited Animal Research Facility (ARF) of the University of New Mexico Health Sciences Center (UNM-HSC) and institutionally approved husbandry conditions and approved breeding protocols were followed (Protocol number 23-201351-B-HSC). M. tuberculosis-infected animals were housed in a separate ABSL3 suite within the UNM HSC ARF facility and all staff followed strict ABSL3, BSL3, and animal protocols approved by the UNM HSC Biosafety Committee and the Institutional Animal Care and Use Committee. The protocol number was 23-201379-HSC. The study was compliant with all relevant ethical guidelines for animal research.

Reviewer #1 (Public review): https://doi.org/10.7554/eLife.100928.3.sa1
Reviewer #2 (Public review): https://doi.org/10.7554/eLife.100928.3.sa2
Reviewer #3 (Public review): https://doi.org/10.7554/eLife.100928.3.sa3
Author response https://doi.org/10.7554/eLife.100928.3.sa4

## Additional files

### Supplementary files
MDAR checklist

## Data availability

All data generated or analyzed during this study are included in the manuscript and supporting files; source data files have been provided for all figures. Raw MS DIA/DDA data have been deposited at the MassIVE proteomics repository MassIVE (MSV000090348) and Proteome Exchange (PXD036850).

The following datasets were generated:

| Author(s) | Year | Dataset title | Dataset URL | Database and Identifier |
|---|---|---|---|---|
| Wang et al | 2023 | Unique position of ATG5 in the atg8ylation cascade provides a switch between autophagy and secretion | https://massive.ucsd.edu/ProteoSAFe/dataset.jsp?accession=MSV000090348 | Massive, MSV000090348 |
| Phinney B | 2023 | Unique position of ATG5 in the atg8ylation cascade provides a switch between autophagy and secretion | https://proteomecentral.proteomexchange.org/cgi/GetDataset?ID=PXD036850 | ProteomeXchange, PXD036850 |

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
