## [Editor Report · eLife Assessment]

Masroor Ahmad Paddar and colleagues reveal noncanonical roles of ATG5 and membrane ATG8ylation in regulating retromer assembly and function. They identify ATG5's unique non-autophagic role and show that CASM partially contributes to these phenotypes. Although the mechanism by which ATG8ylation regulates the retromer remains unclear, the findings provide **important** insights with **solid** supporting evidence.

---

## [Referee Report · Reviewer #1 (Public review)]

Summary:

In this study, Masroor Ahmad Paddar and his/her colleagues explore the noncanonical roles of ATG5 and membrane atg8ylation in regulating retromer assembly and function. They begin by examining the interactomes of ATG5 and expand the scope of these effects to include homeostatic responses to membrane stress and damage.

Strengths:

This study provides novel insights into the noncanonical function of ATG8ylation in endosomal cargo sorting process.

Weaknesses:

The direct mechanism by which ATG8ylation regulates the retromer remains unsolved.

Comments on revisions:

After revision, though the major weakness remains unsolved, other questions have been addressed experimentally or further interpreted.

---

## [Referee Report · Reviewer #2 (Public review)]

Summary:

Padder et al. demonstrates that ATG5 mediates lysosomal repair via the recruitment of the retromer components during LLOMe-induced lysosomal damage and that mAtg8-ylation contributes to retromer-dependent cargo sorting of GLUT1. Although previous studies have suggested that during glucose withdrawal, classical autophagy contributes to retromer-dependent GLUT1 surface trafficking via interactions between LC3A and TBC1D5, the experiments here demonstrate that during basal conditions or lysosomal damage, ATGs that are not involved in mATG8ylation, such as FIP200, are not functionally required for retromer-dependent sorting of GLUT1. Overall, these studies suggest a unique role for ATG5 in the control of retromer function, and that conjugation of ATG8 to single membranes (CASM) is a partial contributors to these phenotypes.

Strengths:

(1) Overall, these studies suggest a unique non-autophagic role for ATG5 in the control of retromer function. They also demonstrate that conjugation of ATG8 to single membranes (CASM) is a partial contributors to these phenotypes. Overall, these data point to a new role for ATG5 and CASM-dependent mATG8ylation in lysosomal membrane repair and trafficking.

(2) Although the studies are overall supportive of the proposed model that the retromer is controlled by CASM-dependent mATG8-ylaytion, it is noteworthy that previous studies of GLUT1 trafficking during glucose withdrawal (Roy et al. Mol Cell, PMID: 28602638) were predominantly conducted in cells lacking ATG5 or ATG7, which would not be able to discriminate between a CASM-dependent vs. canonical autophagy-dependent pathway in the control of GLUT1 sorting. Is the lack of GLUT1 mis-sorting to lysosomes observed in FIP200 and ATG13KO cells also observed during glucose withdrawal? Notably, deficiencies in glycolysis and glucose-dependent growth have been reported in FIP200 deficient fibroblasts (Wei et al. G&D, PMID: 21764854) so there may be difference in regulation dependent on the stress imposed on a cell.

Comments on revisions:

My previous comments have been addressed.

---

## [Referee Report · Reviewer #3 (Public review)]

In this manuscript, Padder et al. used APEX2 proximity labeling to find an interaction between ATG5 and the core components of the Retromer complex, VPS26, VPS29, and VPS35. Further studies revealed that ATG5 KO inhibited the trafficking of GLUT1 to the plasma membrane. They also found that other autophagy genes involved in membrane atg8ylation affected GLUT1 sorting. However, knocking out other essential autophagy genes such as ATG13 and FIP200 did not affect GLUT1 sorting. These findings suggest that ATG5 participates in the function of the Retromer in a noncanonical autophagy manner. Overall, the methods and techniques employed by the authors largely support their conclusions. These findings are intriguing and significant, enriching our understanding of the non-autophagic functions of autophagy proteins and the sorting of GLUT1.

Comments on revisions:

The concerns I raised have all been addressed.

---

## [Author Response]

The following is the authors’ response to the original reviews.

**Public reviews**:
**Reviewer #1 (Public Review):**
Summary:In this study, Masroor Ahmad Paddar and his/her colleagues explore the noncanonical roles of ATG5 and membrane atg8ylation in regulating retromer assembly and function. They begin by examining the interactomes of ATG5 and expand the scope of these effects to include homeostatic responses to membrane stress and damage.Strengths:This study provides novel insights into the noncanonical function of ATG8ylation in endosomal cargo sorting process.Weaknesses:The direct mechanism by which ATG8ylation regulates the retromer remains unsolved.

We agree with the reviewer. We do however show how at least one aspect of atg8ylation contributes to the proper retromer function, which occurs via lysosomal membrane maintenance and repair. Understanding the more direct effects on retromer will require a separate study. We now emphasize this in the revised manuscript (p. 18) and point out the limitations of the present work (p. 18): “One of the limitations of our study is that beyond effects of membrane atg8ylation on quality of lysosomal membrane and its homeostasis there could be more direct effects of membrane modification with mATG8s that still need to be understood”.

**Reviewer #2 (Public Review):**
Summary:Padder et al. demonstrate that ATG5 mediates lysosomal repair via the recruitment of the retromer components during LLOMe-induced lysosomal damage and that mAtg8-ylation contributes to retromer-dependent cargo sorting of GLUT1. Although previous studies have suggested that during glucose withdrawal, classical autophagy contributes to retromer-dependent GLUT1 surface trafficking via interactions between LC3A and TBC1D5, the experiments here demonstrate that during basal conditions or lysosomal damage, ATGs that are not involved in mATG8ylation, such as FIP200, are not functionally required for retromer-dependent sorting of GLUT1. Overall, these studies suggest a unique role for ATG5 in the control of retromer function, and that conjugation of ATG8 to single membranes (CASM) is a partial contributor to these phenotypes.Strengths:(1) Overall, these studies suggest a unique non-autophagic role for ATG5 in the control of retromer function. They also demonstrate that conjugation of ATG8 to single membranes (CASM) is a partial contributor to these phenotypes. Overall, these data point to a new role for ATG5 and CASM-dependent mATG8ylation in lysosomal membrane repair and trafficking.(2) Although the studies are overall supportive of the proposed model that the retromer is controlled by CASM-dependent mATG8-ylaytion, it is noteworthy that previous studies of GLUT1 trafficking during glucose withdrawal (Roy et al. Mol Cell, PMID: 28602638) were predominantly conducted in cells lacking ATG5 or ATG7, which would not be able to discriminate between a CASM-dependent vs. canonical autophagy-dependent pathway in the control of GLUT1 sorting. Is the lack of GLUT1 mis-sorting to lysosomes observed in FIP200 and ATG13KO cells also observed during glucose withdrawal? Notably, deficiencies in glycolysis and glucose-dependent growth have been reported in FIP200 deficient fibroblasts (Wei et al. G&D, PMID: 21764854) so there may be differences in regulation dependent on the stress imposed on a cell.

We thank the reviewer for the overall assessment of the strengths of the study. We have discussed in the manuscript the elegant study by Roy et al., PMID 28602683. To accommodate reviewer’s comment, we have additionally emphasized in the text that our study is focused on basal conditions and conditions that perturb endolysosomal compartments. We agree with the reviewer that under metabolic stress conditions (such as glucose limitation) more complex pathways may be engaged and have acknowledged that in the discussion. We have now included this in the limitations of the study (p. 18): “Another limitation of our study is that we have focused on basal conditions or conditions causing lysosomal damage, whereas metabolic stress including glucose excess or limitation with its multitude of metabolic effects have not been addressed”.

Weaknesses:(1) Additional controls are needed to clarify the role of CASM in the control of retromer function. Because the manuscript proposes both CASM-dependent and independent pathways in the ATG5 mediated regulation of the retromer, it is important to provide robust evidence that CASM is required for retromer-dependent GLUT1 sorting to the plasma membrane vs. lysosome. The experiments with monensin in Fig. 7C-E are consistent with but not unequivocally corroborative of a role for CASM.

We fully agree with the reviewer. In fact, our data with bafilomycin A1 treatment causing GLUT1 miss-sorting show that it is the perturbance of lysosomes and not CASM per se that leads to mis-sorting of GLUT1 (Fig. 7D,E). Note that it has been shown (PMIDs: 28296541, 25484071 and 37796195) that although bafilomycin A1 deacidifies lysosomes it does not induce but instead inhibits CASM. This is because bafilomycin A1 causes dissociation of V1 and V0 sectors of V-ATPase, unlike other CASM-inducing agents which promote V1 V0 association. Complementing this, our data with ATG2AB DKO and ESCRT VPS37A KO (Fig. 8A-F) indicate that the repair of lysosomes is important to keep the retromer machinery functional (as illustrated in Fig. 8G). This may be one of the effector mechanisms downstream of membrane atg8ylation in general and hence also downstream of CASM. We have revised Fig. 7 title to read “Lysosomal perturbations cause GLUT1 mis-sorting” and have explained these relationships in the text (p. 12-13): “Since bafilomycin A1 does not induce CASM but disturbs luminal pH, we conclude that it is the less acidic luminal pH of the endolysosomal organelles, and not CASM, that is sufficient to interfere with the proper sorting of GLUT1.”

Based on the results shown with ATG16KO in Fig 4A-D, rescue experiments of these 16KO cells with WT vs. C-terminal WD40 mutant versions of ATG16 will specifically assess the requirement for CASM and potentially provide more rigorous support for the conclusions drawn.

We have carried out complementation with ATG16L1 WT and its E230 mutant (devoid of WD40 repeats but still capable of canonical autophagy) and placed these data in Fig. 7 (panels I and J) as recommended by the reviewer. This is now described on p. 13 (To additionally test this notion, we compared ATG16L1 full length (ATG16L1FL) and ATG16L1E230 (Rai et al., PMID 30403914) for complementation of the GLUT1 sorting defect in ATG16L1 KO cells (Fig. 7I,J). ATG16L1E230 [Rai, 2019, 30403914] lacks the key domain to carry out CASM via binding to VATPase 29,30 31-33 but retains capacity to carry out atg8ylation. Both ATG16L1FL and ATG16L1E230 complemented mis-sorting of GLUT1 (Fig. 7I,J). Collectively, these data indicate that it is not absence of CASM/VAIL but absence of membrane atg8ylation in general that promotes GLUT1 mis-sorting.).

(2) Also, the role of TBC1D5 should be further clarified. In Fig S7, are there any changes in the interactions between TBC1D5 and VPS35 in response to LLOMe or other agents utilized to induce CASM?

We thank the reviewer for pointing this out. We do have data with VPS35 in co-IPs shown in Fig. S7. There is no change in the amounts of VPS35 or TBC1D5 in GFP-LC3A co-IPs. We now include in Fig. S7 (new panel D) a graph with quantification in the revised manuscript and emphasize this point (p. 12): “However, under CASM-inducing conditions, no changes were detected (Fig. S7B-D) in interactions between TBC1D5 and LC3A or in levels of VPS35 in LC3A co-IP, a proxy for LC3A-TBC1D5-VPS29/retromer association. This suggests that CASM-inducing treatments and additionally bafilomycin A1 do not affect the status of the TBC1D5-Rab7 system”.

Does TBC1D5 loss-of-function modulate the numbers of GLUT1 and Gal3 puncta observed in ATG5 deficient cells in response to LLOMe?

We agree that TBC1D5 is an interesting aspect. However, because TBC1D5 does not change its interactions in the experiments in our study, we consider this topic (i.e. whether TBC1D5 phenocopies VPS35 and ATG5 KOs in its effects on Gal3) to be beyond the scope of the present work. We underscore that LLOMe (lysosomal damage) mis-sorts GLUT1 even without any genetic intervention (e.g., in WT cells in the absence of ATG5 KO; Fig. 7). Thus, in our opinion the effects of TBC1D5 inactivation may be a moot point.

(3) Finally, the studies here are motivated by experiments in Fig. S1 (as well as other studies from the Deretic and Stallings labs) suggesting unique autophagy-independent functions for ATG5 in myeloid cells and neutrophils in susceptibility to *Mycobacterium tuberculosis* infection. However, it is curious that no attempt is made to relate the mechanistic data regarding the retromer or GLUT1 receptor mis-sorting back to the infectious models. Do myeloid cells or neutrophils lacking ATG5 have deficiencies in glucose uptake or GLUT1 cell surface levels?

Reviewer’s point is well taken. Glucose uptake, its metabolism, and diabetes underly resurgence in TB in certain populations and are important factors in a range of other diseases. This was alluded to in our discussion (lines 461-469). However, these are complex topics for future studies. We have now expanded this section of the discussion (p. 18): “In the context of tuberculosis, diabetes, which includes glucose dysregulation, is associated with increased incidence of active disease and adverse outcomes” (Dheda et al., ,PMID: 26377143; Dooley, et al., PMID:19926034).

**Reviewer #3 (Public Review):**
In this manuscript, Padder et al. used APEX2 proximity labeling to find an interaction between ATG5 and the core components of the Retromer complex, VPS26, VPS29, and VPS35. Further studies revealed that ATG5 KO inhibited the trafficking of GLUT1 to the plasma membrane. They also found that other autophagy genes involved in membrane atg8ylation affected GLUT1 sorting. However, knocking out other essential autophagy genes such as ATG13 and FIP200 did not affect GLUT1 sorting. These findings suggest that ATG5 participates in the function of the Retromer in a noncanonical autophagy manner. Overall, the methods and techniques employed by the authors largely support their conclusions. These findings are intriguing and significant, enriching our understanding of the non-autophagic functions of autophagy proteins and the sorting of GLUT1.Nevertheless, there are several issues that the authors need to address to further clarify their conclusions.(1) The authors confirmed the interaction between Atg5 and the Retromer complex through Co-IP experiments. Is the interaction between Atg5 and the Retromer direct? If it is direct, which Retromer complex protein regulates the interaction with Atg5? Additionally, does ATG5 K130R mutant enhance its interaction with the Retromer?

AlphaFold modeling in the initial submission of our study to eLife (absent from the current version) suggested the possibility of a direct interaction between ATG5 and VPS35 with ATG12—ATG5 complex facing outwards, in which case K130R would not matter. However, mutational experiments in putative contact residues did not alter association in co-IPs. So either ATG5 interacts with other retromer subunits or more likely is in a larger protein complex containing retromer. It will take a separate study to dissect associations and find direct interaction partners.

(2) To more directly elucidate how ATG5 regulates Retromer function by interacting with the Retromer and participates in the trafficking of GLUT1 to the plasma membrane, the authors should identify which region or crucial amino acid residues of ATG5 regulate its interaction with the Retromer. Additionally, they should test whether mutations in ATG5 that disrupt its interaction with the Retromer affect Retromer function (such as participating in the trafficking of GLUT1 to the plasma membrane) and whether they affect Atg8ylation. They also need to assess whether these mutations influence canonical autophagy and lysosomal sensitivity to damage.

Please see the response to point 1.

**Recommendations for the authors.**

**Reviewer #1 (Recommendations For The Authors):**
While most data are solid and convincing, the following questions need to be addressed before publication:Major Concerns:(1) Examining only one cargo (GLUT1) is insufficient to reflect the retromer's function comprehensively. At least two additional cargoes should be analyzed to observe the phenotypes more accurately.

We agree that having another retromer cargo (in addition to GLUT1) would be of interest. We point out that our data also show mis-sorting of SNX27 to lysosomes (Fig. 3H, quantifications in Fig. 3I). SNX27 in turn sorts nearly 80 ion channels, signaling receptors, and other nutrient transporters. Which of the 80 cargos to prioritize and check (the expectation is that all 80 might be missorted given that they need SNX27)? We have instead tested MPR, a SNX27-independent cargo. We now include data on effects of ATG5 knockout on CI-MPR (Fig. S9A-F). This is described in the text (p. 14); “Effect of ATG5 knockout on MPR sorting

We tested whether ATG5 affects cation-independent mannose 6-phosphate receptor (CI-MPR). For this, we employed the previously developed methods (Fig. S9A) of monitoring retrograde trafficking of CI-MPR from the plasma membrane to the TGN 70,118-121. In the majority of such studies, CI-MPR antibody is allowed to bind to the extracellular domain of CI-MPR at the plasma membrane and its localization dynamics following endocytosis serves as a proxy for trafficking of CI-MPR. We used ATG5 KOs in HeLa and Huh7 cells and quantified by HCM retrograde trafficking to TGN of antibody-labeled CI-MPR at the cell surface, after being taken up by endocytosis and allowed to undergo intracellular sorting, followed by fixation and staining with TGN46 antibody. There was a minor but statistically significant reduction in CIMPR overlap with TGN46 in HeLaATG5-KO that was comparable to the reduction in HeLa cells when

VPS35 was depleted by CRISPR (HeLaVPS35-KO) (Fig. S9B,C). Morphologically, endocytosed Ab-CI-

MPR appeared dispersed in both HeLaATG5-KO and HeLaVPS35-KO cells relative to HeLaWT cells (Fig. S9D). Similar HCM results were obtained with Huh7 cells (WT vs. ATG5KO; Fig. S9E,F). We interpret these data as evidence of indirect action of ATG5 KO on CI-MPR sorting via membrane homeostasis, although we cannot exclude a direct sorting role via retromer. We favor the former interpretation based on the strength of the effect and the controversial nature of retromer engagement in sorting of CI-MPR (57,70,75,98,120).

(2) The evidence from Alphafold predictions is weak. The direct interaction of ATG5 with retromer subunits should be tested.

Please see the above response to Reviewer 3.

In addition, does retromer also interact with ATG16L1 similarly to the phenomenon in VAIL?

We fully agree with the reviewer that finding the direct interacting partners between retromer and membrane atg8ylation machinery is an important direction as in our opinion it would expand the repertoire of E3 ligases and its adaptors. However, given the complexity and variety of possibilities, we believe that this is a topic for a future study.

(3) In Line 166, Figures 2C and 2D, the Gal3 phenotype does not seem to be well complemented by VPS35.

We have adjusted the text to acknowledge incomplete complementation (p.7).

(4) In Figures 3 and 4, the authors show that KO of membrane atg8ylation machineries and ATG8-Hexa KO affects the localization of retromer cargo GLUT1 and SNX27. However, the mechanism by which membrane ATG8ylation affects retromer remains unresolved.Additionally, are other retromer subunits' locations are also affected, if so, how are they impacted? At least a speculative explanation should be provided.

Following reviewers request, we now state on p. 19 that “one of the limitations of our study is that beyond effects of membrane atg8ylation on quality of lysosomal membrane and its homeostasis there could be more direct effects of membrane modification with mATG8s on retromer that still need to be understood”.

(5) In Figure 3, endogenous IP results are required to examine the interaction of ATG5 with retromer if suitable retromer antibodies for IP are available.

Endogenous IPs are given in Fig. 1. We have modified text on p. 8 to clarify this.

(6) In Figure 4, ATG8 Hexa KO, and triple KO of LC3s or GABARAPs all increase the localization of GLUT1 on lysosomes. It seems redundant for ATG8 family proteins here.

Can any individual member of the ATG8 family rescue this phenotype?

If the intent of such complementation analysis is to identify a specific mATG8 responsible for the observed effects, this is already pre-empted by the fact that TKOs also have a similar effect as HEXA mutants (i.e. loss of at least two of mATG8s is enough to cause the phenotype). We now discuss this in the text (p. 10): “Thus, at least two mATG8s, each one from two different mATG8 subclasses (LC3s and GABARAPs) or the entire membrane atg8ylation machinery was engaged in and required for proper GLUT-1 sorting”.

(7) In Figure 5, knockdown of ATG5 in FIP200 KO cells inhibited GLUT1 sorting from endosomes, leading to its trafficking to lysosomes. However, it is known that very little remnant ATG5 in ATG5 KD cells is enough to support ATG8 lipidation. Therefore, it is essential to repeat this experiment using ATG5/FIP200 double KO or ATG5 KO combined with an autophagy inhibitor.

We point out to this limitation in the text (p. 11): “….we knocked down ATG5 in FIP200 KO cells (Fig. S5D) and found that GLUT1 puncta and GLUT1+LAMP2+ profiles increased even in the FIP200 KO background with the effects nearing those of VPS35 knockout (Figs. 5D-F and S5C), with the difference between VPS35 KO and ATG5 KD attributable to any residual ATG5 levels in cells subjected to siRNA knockdowns”.

(8) In Figure 7, the authors show that the induction of CASM inhibited GLUT1 sorting from endosomes. However, ATG5 KO, which abolishes membrane ATG8ylation, also inhibits GLUT1 sorting. This seems paradoxical and requires a reasonable explanation or discussion.

We understand reviewer’s comment. The answer to this paradox is that it is actually the lysosomal damage that causes GLUT1 mis-sorting and not CASM. Membrane atg8ylation, such as CASM and probably other processes given that involvement of both ATG2 and ESCRTs (Fig. 8) counteracts the damage and works in the direction of restoring/maintaining proper retromer-dependent sorting. This is now explained better in the text, and have revised the title of Fig. 7 to read “Lysosomal damage causes GLUT1 mis-sorting”. Our data with bafilomycin A1 show that it is the perturbance of lysosomes (not CASM per se) that leads to mis-sorting of GLUT1 (Fig. 7D,E), and our data with ATG2AB DKO and ESCRT (VPS37A) KO (Fig. 8A-F) indicate that repair of lysosomes is important to keep the retromer working machinery functional (as illustrated in Fig. 8G), which may be one of the effector mechanisms downstream of membrane atg8ylation in general (and hence also of CASM).

(9) The immuno-staining results for Figures 7F and 7G are lacking.

We now provide the requested images.

(10) In Figure 8D, the quality of the image for VPS37 KO cells treated with LLOME is not sufficient to show increased colocalization between GLUT1 and LAMP2.

We now provide a different example image. We note that these are epiflorescent HCM images

Minor Concerns:(1) It would be better to distinguish the function of the membrane ATG8ylation machinery (i.e., ATG5) from the function of membrane ATG8ylation in the description. No ATG8ylation-deficient mutants were used in this study.

We have used atg8ylation mutants (e.g. KOs in ATG3, ATG5, ATG7, and ATG16L1). We now emphasize this better in the text (p. 10).

(2) In Figure 2D, a green box appears there by incident.

This has been fixed.

(3) In Figure 3A, the conjugate for ATG5-ATG12 is absent in the gel for IB: ATG5.

The ATG5 antibody used in Fig. 3A recognizes primarily the conjugated form of ATG5. This is now clarified in the figure legend.

(4) Figure 5G is missing in the manuscript.

Fig 5G is now mentioned in the text. Thank you.

(5) The gRNA sequence information for FIP200 KO is missing in the Methods section.

Reference(s) to the already published gRNA sequence are in the manuscript.

(6) Suggest moving the last paragraph in Result section to Discussion section.

We kept this single-paragraph section in Results as it contains actual data.

**Reviewer #2 (Recommendations For The Authors):**
(1) It is unclear why the rescue of VPS35KO cells in Fig 1C-D is so modest.

Complementation data depend on transfection efficiency and some variability is to be expected.

**Reviewer #3 (Recommendations For The Authors):**
(1) Figures 2A, 2C, 2E, and 2G lack scale bars. Figure 2D has a small square above the y axis.

Relative scale bars are now included.

(2) Figures S3B, S3D, and S3F lack scale bars.

Relative scale bars are now included.